

# Application of Open Path Fourier Transform Infrared Spectroscopy (OP-FTIR) to Measure Greenhouse Gas Concentrations from Agricultural Soils

Cheng-Hsien Lin[1], Cliff T. Johnston[1], Richard H. Grant[1], and Albert J. Heber[2]

[1]Department of Agronomy, Purdue University, West Lafayette, IN 47907, United States
[2]Department of Agricultural and Biological Engineering, Purdue University, West Lafayette, IN 47907, United States

*Correspondence to*: Cheng-Hsien Lin (lin471@purdue.edu)

**Abstract.** Open-path Fourier transform infrared spectroscopy (OP-FTIR) has often been used to measure hazardous or trace

gases from the "hot" point sources (e.g., volcano, industrial or agricultural facilities) but seldom used in the field-scale

source areas, such as soil emissions. OP-FTIR, the close-path mid-IR laser-based $N_2O$, and the nondispersive-IR $CO_2$

analyzers were used to measure the concentrations of greenhouse gases (e.g., $N_2O$ and $CO_2$) emitted from agricultural soils

over a period of 9-19[th] June in 2014. We developed a quantitative method of $N_2O/CO_2$ analysis that minimized the

interferences from diurnal changes of humidity and temperature in order to measure $N_2O/CO_2$ concentrations accurately.

Two chemometric multivariate models were developed, a classical least squares (CLS) and a partial least squares (PLS),

respectively. This study evaluated different methods to generate the single beam background spectra, and different spectral

regions to determine $N_2O/CO_2$ concentrations from OP-FTIR spectra. A standard extractive method was used to measure the

'actual' path-averaged concentrations along an OP-FTIR optical path in situ, as a benchmark to assess the feasibilities of

these quantitative methods. Within the absolute humidity of 5,000-20,000 ppmv and the temperature of 10-35 ℃, we found

that the CLS model underestimated $N_2O$ concentrations (Bias = -4.9±3.1 %) calculated from OP-FTIR spectra, and the PLS

model improved the accuracy of the calculated $N_2O$ (Bias = 1.4±2.3 %). The bias of the calculated $CO_2$ was -1.0±2.8 %

using the CLS model. These methods suggested that the changed ambient factors potentially led to biases in $N_2O/CO_2$

estimations from OP-FTIR spectra, and may help the OP-FTIR user to escape from the dependency of extractive methods

used to calibrate the concentration determined by OP-FTIR.

## 1 Introduction

Agriculture contributes a substantial amount of greenhouse gas (GHG) emissions (e.g. $N_2O$, $CO_2$, and $CH_4$) to the global

GHG budget (IPCC, 2007; Cole et al., 1997; Smith et al., 2008). Among these gases, $N_2O$ is mainly emitted from

agricultural soils, accounting for 38% of the global anthropogenic non-$CO_2$ GHG emissions from agricultural activities

(Smith et al., 2007; US-EPA, 2006). $N_2O$ is produced from biological reactions that transform available N in soils via

microbial nitrification and denitrification (Mosier et al., 2004). Taking into account that the global warming potential value





of $N_2O$ is 310, it is estimated that overall GHG emission from soils (based on $CO_2$ equivalents) is approximate 2500 $MtCO_2$-eq $yr^{-1}$. A significant fraction of soil $N_2O$ emissions results from the use of nitrogen fertilizers in agricultural soils. In addition to contributing to the overall GHG burden of the atmosphere, $N_2O$ emissions also represent a direct loss of the N applied to the field, contributing to the decreased nitrogen use efficiency (NUE) (Eichner, 1990; Ryden and Lund, 1980;

Bremner et al., 1981). Also, soils play the role of a sink or a source for the atmospheric $CO_2$ (Paustian et al., 1997; Smith et al., 2008). Changing land use, especially agricultural-related uses such as tile drained and tilled managements, and agricultural lime application (e,g., $CaCO_3$ and $MgCa(CO_3)_2$) potentially become a large source of $CO_2$ released to the atmosphere via microbial decomposition of soil organic carbon (Smith, 2004; IPCC, 2007; Cole et al., 1997, West et al., 2005).

Chamber measurements have been the most common method to measure GHG emissions from soils (Denmead, 2008; Rochette and Eriksen-Hamel, 2008). Chamber measurements, however, are subject to significant limitations that lead to uncertainties and biases in the estimated GHG emissions. For instance, because chambers have a small footprint (~0.5 $m^2$) and generally wide sampling intervals (usually once a week), they are poorly suited to study the spatial and temporal

variability of GHGs from agricultural soils (Laville et al., 1999; Rowlings et al., 2012; Schelde et al., 2012). Also, the increased wind turbulence substantially induced more gas transportation from soils to the atmosphere. Chamber methods unlikely consider the wind-induced effect into account, so likely resulting in underestimations for gas measurements (Denmead and Reicoshky, 2003; Poulsen et al., 2017; Pourbakhtiar et al., 2017).

Open-path Fourier transform infrared spectroscopy (OP-FTIR) is a non-intrusive sensing approach and capable of detecting multiple components simultaneously, acquiring real-time data at a relatively high temporal resolution (second to minutes), and providing path-averaged gas concentrations (Russwurm et al., 1991). OP-FTIR has been applied to measure atmospheric gases since the 1970s (e.g., hazardous air pollutants, fugitive volatile organic compounds (VOCs), and trace gases) (Herget and Brasher, 1980; Gosz et al., 1988; Russwurm et al., 1991; Bacsik et al., 2006; Briz et al., 2007; Lin et al., 2008). More

recently, OP-FTIR has been increasingly applied to measure GHGs or other trace gases in agriculture, mostly in animal facilities (e.g., $N_2O$, $CO_2$, $CH_4$, and $NH_3$ from the swine or dairy production facility) (Childers et al., 2001; Loh et al., 2008; Bjorneberg et al., 2009; Barrancos, 2013; Naylor et al., 2016). Only a few studies, however, implemented OP-FTIR to measure gas emissions from vegetable production fields or fertilized soils (Bai et al., 2014 and 2018; Ni et al., 2015). Integrating OP-FTIR with the micrometeorological techniques (e.g., flux gradient (FG) or backward Lagrangian stochastic

dispersion (bLS) methods) can measure gas fluxes from the field-scale source of interest with high temporal and spatial representatives that are less prone to artifacts induced by point-based sampling (Flesch et al., 2004 and 2016; Bai et al., 2014 and 2018; Ni et al., 2015). Moreover, the OP-FTIR combined with a scanning system can potentially be applied to horizontally or vertically survey numerous fields of interest and measure their gas emissions simultaneously (Flesch et al., 2016).



Despite these advantages, OP-FTIR also faces a number of challenges. In order to resolve the spectral features of GHGs, high spectral resolution ($< 0.5$ cm$^{-1}$) is required to resolve the rotation-vibrational absorption bands of the GHGs of interest (Griffiths and de Haseth, 2007). Calculating concentrations from FTIR spectra requires both a 'sample' single beam spectrum and a reference/background spectrum that does not contain spectral contributions from GHGs of interest, which is

not possible at the field scale (e.g., evacuation of the field); thus, mathematical methods have been developed which strip the spectral bands from a 'sample' single beam spectrum. This challenge requires the use of instrumental- or spectral-processing methods to create a background spectrum, and these methods are subject to biases to determine GHG concentrations (Griffiths and de Haseth, 2007; Russwurm and Childers, 1996). Furthermore, the atmosphere contains the high concentration of water vapour that interferes with the detection and quantification of GHGs of interest (Russwurm and Childers, 1996;

Horrocks et al., 2001; Briz et al., 2007; Smith et al., 2011). These challenges of data processes and the interferences from water vapour likely introduced biases and uncertainties in the GHG quantification. Using the error-prone concentration in flux prediction models (micrometeorological techniques) possibly leads to unknown uncertainties in the estimated gas fluxes. Thus, it is essential to develop a comprehensive quantitative method to improve and assure the quality of gas quantification using OP-FTIR.

Testing the feasibility of quantitative methods and qualities (accuracy and precision) of OP-FTIR is challenging because a trustable reference is required to validate the FTIR-derived concentrations. One of the most common approaches was to position a gas cell filled with known gas concentrations of interest in the optical path and test their quantitative methods (Russwurm et al., 1991; Horrocks et al., 2001; Smith et al., 2011). This approach, however, somewhat controlled the

environment and neglected the effect of the ambient interferences, such as water vapour, on the quality of gas quantification. The alternative approach is to compare the derived concentrations with ambient concentrations. The ambient concentration of a gas of interest can be determined by averaging the global background concentrations (e.g. $N_2O$~310 ppbv or $CO_2$~400 ppmv) or measured from the gas samples that were collected along OP-FTIR path and analysed their concentrations using Lab-based gas chromatography (GC) (ASTM, 2013; Childers et al., 1995; Kelliher et al., 2002; Bai et al., 2014). The

experimental designs of these assessment approaches, either the point sampled setup or the low sampled frequency or both, became the major problem for cross-validating their OP-FTIR quantitative methods. Since the ambient concentrations likely fluctuate from place to place (e.g., different land uses) and in different timing (e.g., diurnal or seasonal variation), the spatial and temporal variations of the ambient concentration were potentially misconceived as "bias" in gas quantification. Up to now, only three studies continuously measured the real-time ambient concentration to logically cross-validate their

quantitative methods as well as data qualities under the fluctuated environmental factors (e.g., the dynamic water vapour), but none of the prior studies assess the methodology for quantifying $N_2O$ concentrations (Briz et al., 2007; Frey et al., 2015; Reiche et al., 2014).



Therefore, the objectives of this study were to 1) develop a long-path gas sampling system that can continuously collect numerous gas samples along an optic path of OP-FTIR simultaneously and measure the path-averaged concentrations to evaluate the quantitative qualities of $N_2O/CO_2$ concentrations that were derived from OP-FTIR spectra, and 2) optimize the quantitative method, including post-data processing, analytical window selections, and chemometric multivariate algorithms

(i.e., classical least square and partial least square), that was less sensitive to the interference of ambient factors (i.e., humidity and temperature) and capable of determining $N_2O$ and $CO_2$ concentrations accurately.

## 2 Materials and experimental methods

### 2.1 Site description

This study was conducted at the Purdue University Agronomy Center for Research and Extension near West Lafayette,
Indiana, the United States (86°56´ W, 40°49´ N, elevation 215 m). The experimental site was located between two fields (~3.5 hectares in each field) with a continuous corn system since 2013. Gas measurements began just after the anhydrous ammonia application with total N rate of 220 kg $NH_3$-N ha$^{-1}$ on June 9th and ended on June 19th. The soils were classified as Drummer silty clay loam (fine-silty, mixed, mesic Typic Endoaquoll) with the bulk density of 1.6 g·m$^{-3}$, organic matter of 3.4 %, soil pH of 6.0, and cation exchange capacity of 23 cmol$_c$·kg$^{-1}$ (0-20 cm). During June 9-19th, the cumulative
precipitation was 57 mm, and the average soil temperature and moisture from the depth of 0-10 cm were 23(±3) °C and 0.32(±0.06) cm$^3$·cm$^{-3}$, respectively, which were determined by the on-site weather station.

### 2.2 Instrumentation setup (Fig.1)

The spectrometer was used a MIDAC Corporation monostatic open path FTIR air monitoring system (MIDAC Model2501-C, MIDAC Corporation, Irvine, CA). This instrument included the IR source, interferometer, transmitting/receiving
telescope, mercury cadmium telluride (MCT) detector and ZnSe optics. A mid-IR beam in the spectrometer passed through the atmosphere along an optical path and returned to the telescope after reflection from a retro-reflector to collect spectra that included the information of the gas of interest. A cube-corner retroreflector with 26 cubes was mounted on a retractable tripod with 150 m physical path length from the telescope, corresponding to an optical path length of 300 m.

Ambient concentrations of $N_2O$ and $CO_2$ were also determined independently to assess the bias and precision. A difference frequency generation (DFG) mid-IR laser-based $N_2O/H_2O$ analyzer (IRIS 4600, Thermo Fisher Scientific Inc., MA) and the non-dispersive infrared (NDIR) spectrometer $CO_2/H_2O$ gas analyzer (LI-840, LI-COR Inc., NE) were used to measure $N_2O$ and $CO_2$ concentrations of the sampled gases from a synthetic open path gas sampling system (S-OPS). The DFG laser-based $N_2O$ analyzer determined $N_2O$ concentrations in the mid-infrared wavelength with high precision of < 0.15 ppbv (1σ,
3 minute averaging). An NDIR $CO_2$ analyzer provided the high accuracy (< 1.5 % of reading) and low noise (< 1.0 ppmv) to determine $CO_2$ concentrations using single path, dual wavelength, and infrared detections system.





A 50 m long S-OPS combined with a gas sampling system (GSS) was used to collect gas samples along an optical path of OP-FTIR. An S-OPS consisted of 3/8 inch diameter Teflon® tubes and ten inlets fitted with one μm Teflon® filters. The flow rate of each inlet was adjusted by critical orifices in the inlets to 0.7 L·min⁻¹ (±10 %). Gas samples were drawn through

an S-OPS line by a sampling pump in this GSS system at approximately 7 L·min⁻¹ and collected into Teflon® ambient pressure chamber. Then, $N_2O$ and $CO_2$ analyzers drew path-integrated air samples from the ambient pressure chamber (Heber et al., 2006), and the "actual" path-averaged concentrations of $N_2O$ and $CO_2$ along the OP-FTIR path, which was used to benchmark $N_2O/CO_2$ concentrations calculated from the OP-FTIR spectrum. Temperature, relative humidity, and pressure in the ambient pressure chamber were also recorded every 30 seconds to monitor the functionality of the GSS.

Meteorological measurements of air temperature, relative humidity (HMP45C, Vaisala Oyj, Helsinki, Finland), and barometric pressure (278, Setra, Inc., Boxborough, MA) were at 1.5 m height of the mast located next to the S-OPS. The meteorological data were collected by a data logger (Model CR1000, Campbell Scientific, Logan, Utah), and averaged every 10 minutes. Wind speed and direction were acquired from a 3D sonic anemometer (Model 81000, RM Young Inc., Traverse

City, MI) mounted at 2.5 m on the meteorological mast and recorded at 5 min intervals. The recorded data were telemetered to the on-site instrumentation trailer.

**2.3 General overview of $N_2O/CO_2$/water vapour concentrations, and air temperature**

Figure 2 shows the 30-min averaged concentrations of ambient $N_2O$ and $CO_2$ measured from the S-OPS, water vapour content and air temperature from the meteorological station during the period of 9-19[th] June in 2014. During this time

interval, 877 valid OP-FTIR spectra were collected with known concentrations of $N_2O$, $CO_2$, water vapour, and air temperature. To avoid the non-linear response of absorbance to the wide range of gas concentrations (Lamp et al., 1997), we selected ninety spectra containing 338±0.5 ppbv $N_2O$ and ninety-three spectra containing 400±5 ppmv $CO_2$ which were measured from the S-OPS. These group of spectra covered broad ranges of water vapour content and air temperature. $N_2O$ and $CO_2$ concentrations were calculated from these selected spectra using different quantitative methods.

**2.4 OP-FTIR data acquisition and QA/QC procedure**

Each sampled spectrum was acquired by co-adding 64 single-sided interferograms (IFGs) at a nominal resolution of 0.5 cm⁻¹, which accounted for 32,000 data points were collected with the interval of 0.241 wavenumbers between data points, using the AutoQuant Pro4.0 software package (MIDAC Corporation, Irvine, CA). IFGs were converted to single beam (SB) spectra using a zero-filling factor of 1, triangular apodization, and Mertz phase correction. A stray light SB spectrum was

also acquired by pointing the transmitting/receiving telescope away from the retroreflector at the beginning of the experiment every day using the same parameters (Russwurm and Childers, 1996). Each sampled SB spectrum was stray light





corrected by subtracting the stray light SB spectrum from the sampled SB spectrum before converting to the absorbance spectrum.

The IFGs and corresponding SB spectra were influenced by ambient factors that included wind-derived vibrations, scintillation induced by air mixing, water vapour content, dust accumulation and condensation on the retro-reflector. Criteria of quality assurance were based on the inspection of the IFG and the SB spectrum, which are followed the standard guideline in the MIDAC instrumentation manual and the FTIR open-path monitoring guidance documents (Russwurm and Childers, 1996) with the supplement criteria published by Childers et al. (2001b) and Shao et al. (2007) to acquire the high-quality spectrum. The maximum and minimum of the centerburst of the IFG were controlled between approximately 0.61-1.14 Volts based on the physical path length of 150 m. IFG centerburst signals > 2.25 Volts were rejected to avoid a non-linear response of the MCT detector.

## 2.5 Spectral analyses

### 2.5.1 An absorbance spectrum converted from a single beam (SB) spectrum

In order to calculate a concentration for a given solute, a stray-light corrected SB spectrum is ratioed against an SB background spectrum (GHGs-free) to produce an absorbance spectrum from which the gas concentration is determined using the Beer-Lambert law. As discussed earlier, OP-FTIR measurements do not permit the collection of a background spectrum that is 'free' of GHGs. Two different approaches were used in this study to overcome this constraint. Both methods required a 'normal' SB spectrum corresponding to the path length of interest that was then mathematically manipulated to produce a background spectrum. A representative field SB spectrum and the regions of interest for each of GHGs are shown in Figure 3(a). For the "zapped" background method, a background (zap-bkg) was obtained by drawing a straight line between two selected points which removed, or 'zapped,' any spectral contributions below the line using OMINC Macro Basic 8.0 commercial software (Thermo Fisher Scientific, Inc.). This is illustrated for the $N_2O$ region of interest in Figure 3(b), with the two points and the line labeled as 'zapped' background. For the zap-bkg method, we selected one quality SB spectrum every day to create a zap-bkg, and all of the sampled SB spectra collected from one day were converted to absorbance spectra using this zap-bkg. Another method, referred to as the 'synthetic' background method, was generated from this same original SB spectrum using IMACC software (Industrial Monitoring and Control Corp., Texas). In this case, numerous points in the 'non-absorbing' region of the SB spectrum were selected as 'base points,' and a high-order fitting function was used to construct a background spectrum. An example in the $N_2O$ region is illustrated in Figure 3(b) and is labeled 'synthetic' background (syn-bkg). The mathematically manipulated SB spectra were used as background files to convert the sampled SB spectra into absorbance spectra (Fig.3c and 3d). For the syn-bkg method, all data points were stored as one data file, and this file was applied to each sampled SB spectrum to create its syn-bkg. Since the selected points determined the curvature of the syn-bkg SB spectrum, it is critical to choose the data points that do not introduce any distortion (e.g. artificial dips and





peaks) into the curvature of the syn-bkg. In general, we avoided selecting data points within the absorption feature of interest (e.g. 2170-2224 cm$^{-1}$ for $N_2O$ analysis), and an adequate number of data points was used to fit the curvature of the SB spectrum as long as we can produce a smooth function (Russwurm and Childers, 1996). Adding too many data points may lead to the artificial distortion in a syn-bkg. Because the syn-bkg is one of the recommended methods used in the spectral

analysis (ASTM, 2013), this method was used to assess the feasibility of the zap-bkg method.

### 2.5.2 Gas quantifications: Multivariate models and spectral window selections

Based on Beer-Lambert law, we used reference spectra to predict gas concentrations from field absorbance spectra. In this study, we used classical least squares (CLS) and partial least squares (PLS) regressions to calculate $N_2O$ and $CO_2$ concentrations. The details of these two methods are described as follows.

- CLS prediction model: Each of the reference spectra used in the CLS model only contained one gas component (e.g. $N_2O$, $CO_2$, or water vapour), and these reference spectra were generated from the high-resolution transmission molecular absorption (HITRAN) database (Rothman et al., 2005). The CLS model (AutoQuant Pro4.0) predicted gas concentrations from the field absorbance spectra converted using the zap-bkg method. In addition, CLS spectra were

also calculated using the IMACC software to predict gas concentrations from the spectra converted by the syn-bkg method. The non-linear function between the actual and predicted gas concentrations of the reference spectra was selected in the CLS model in both quantitative packages.

- PLS prediction model: Each of the reference spectra used in the PLS model had multiple gas components (e.g. an

$N_2O/H_2O$ mixing spectrum). Gas samples were delivered to a multi-pass gas cell (White cell) with an optical path length of 33 m (Model MARS-8L/40L, Gemini Scientific Instruments, CA). Spectra were collected by a laboratory-based FTIR spectrometer (Nexus 670, Thermo Electron Corporation, Palatine, IL), including globar IR source, KBr beam splitter, and a mercury cadmium telluride High D* (MCT-High D*) detector. The FTIR was purged with dry air (-20 ℃ dew point) produced from a zero air generator (Model 701H, Teledyne, CA). $N_2O$ was diluted with ultra-pure nitrogen

gas using a diluter (Series 4040, Environics Inc, CT), and the water vapour content was controlled by a Nafion tube contained within a sealing container of the saturated water vapour. Temperature and humidity were monitored using a Vaisala model HMT 330 humidity and temperature transmitter (Vaisala Oyj, Helsinki, Finland). $N_2O$ concentrations were diluted from 30 ppmv to 0.3, 0.4, 0.5, 0.6 and 0.7 ppmv mixing with the relative humidity of 20, 40, 60, and 80 % at 303 K. Spectra were acquired at 0.5 cm$^{-1}$ resolution, averaged from 64 sample scans with triangular apodization. A

total of 60 spectra of $N_2O/H_2O$ mixing gases were used to build the PLS model using TQ Analyst software Version 8.0 (Thermo Fisher Scientific, Inc.) In order to avoid over-fitting the models, the optimum of factors used in PLS models were determined by cross-validation and justified by the prediction residual error sum of squares (PRESS) function. The




correlation between the known and the PLS-predicted concentrations was used to quantify $N_2O$ from the field absorbance spectrum converted by syn-bkg within given spectral windows.

- Spectral window selections: The window selection (Fig. 4) was critical because of the interferences of water vapour.
While a broader window contained more information of the gas of interest and potentially improved the spectral fit between the modeled and sampled spectra and the quantitative accuracy, it also included more features of water vapour and led to biases in gas quantifications. On the other hand, a narrow window can minimize the interfering effect of the uninteresting gases but may reduce the spectral information of the targeted gas which led to biases in gas calculations (e.g. underestimation for the gas quantification). The window used for $N_2O$ quantifications was from 2130 to 2224 cm$^{-1}$
that mainly includes the absorbance features of $N_2O$ (P-branch) and water vapour, and we also selected different regions for calculating $N_2O$ concentrations. For $CO_2$, the spectral windows of 2070-2085 cm$^{-1}$ and 722-800 cm$^{-1}$ (not shown) contains the features of $CO_2$ and water vapour (Rothman et al., 2005). We selected the multi-windows to calculate $CO_2$ concentrations and assessed the effect of water vapour on gas predictions.

## 2.6 The accuracy of the FTIR-calculated concentration and statistical analysis

Bias, the relative error between the S-OPS and OP-FTIR measured $N_2O/CO_2$, indicated the accuracy of the calculated $N_2O$ and $CO_2$ concentrations using different spectral analyses (i.e. background types, multivariate models, and spectral windows) and can be calculated following Eq. (1):

$$Bias = \frac{(x_i - x_t)}{x_t} \times 100\% \tag{1}$$

, where $x_i$ is the $N_2O$ or $CO_2$ concentration calculated from the OP-FTIR spectrum, and $x_t$ is the known $N_2O$ or $CO_2$
concentration measured from the S-OPS. The calculated concentrations were statistically analysed by ANOVA procedures and protected least significant difference (LSD) ($\alpha$=0.05) (SAS 9.3; SAS Institute Inc., 2012).

## 3 Results and discussion

### 3.1 Quantitative methods (SB backgrounds, spectral windows, and multivariate models)

Both SB background methods, zap-bkg and syn-bkg, respectively, were used to convert the sampled SB spectra to
absorbance spectra for gas quantifications. Various windows were used to calculate gas concentrations from the field-measured OP-FTIR spectrum using CLS and PLS models. A series of OP-FTIR spectra acquired from broad ranges of humility (i.e., 5,000-20,000 ppmv water vapour) and temperature (10-35 °C) were used to calculate $N_2O$ and $CO_2$ concentrations. Within these ranges, the mean bias (%) indicated the accuracy of the calculated $N_2O$ or $CO_2$ and the standard deviation (SD) referred to the sensitivity of the quantitative method to the changed water vapour and air temperature.



## 3.2 Nitrous oxide (338 ppbv)

Generally, the accuracy of the calculated $N_2O$ concentration (mean bias) was improved by narrowing the spectral window because of the lessened water absorption features. In the CLS model, the broadest window ($W_N1$: 2170-2223.7 $cm^{-1}$) (Fig. 4) led to an underestimate of 10.7($\pm$2.3) % in $N_2O$ calculations from the absorbance spectra that were converted by zap-bkg,

and this bias can be reduced using $W_N2$ (2188.5-2223.7 $cm^{-1}$) (i.e. 9.1$\pm$2.5 % underestimate). Likewise, $W_N1$ led to an underestimate of 8.1($\pm$2.6) % in $N_2O$ calculations using syn-bkg, and this bias was reduced using $W_N3$ (2215.8-2223.7 + 2188.5-2204.1 $cm^{-1}$) (i.e. 5.6$\pm$2.6 % underestimate). Although narrowing the window mitigated the features as well as interferences of water vapour, it also lost the information of $N_2O$ and potentially resulted in a great bias to predict $N_2O$ concentrations if the analytical window was over confined. The most confined window ($W_N4$: 2188.5-2204.1 $cm^{-1}$) used in

the CLS model gave rise to greater biases in both zap- and syn-bkg procedures. Beside $W_N1$ (2170-2223.7 $cm^{-1}$), the P-branch feature of $N_2O$ extended from 2130 to 2223.7 $cm^{-1}$, and we also used this entire region to calculate the $N_2O$ concentration. In CLS model, the window of 2130-2223.7 $cm^{-1}$ showed the minimum mean bias of -0.4% of the calculated $N_2O$ concentrations using syn-bkg (data not shown); however, this window was sensitive to a water vapour interference and led to the highest variability in $N_2O$ estimations (i.e., -0.4$\pm$5.3 %). As previously mentioned, it was crucial to generate a

reasonable background for the spectral analysis. The $N_2O$ concentration calculated from the absorbance converted by syn-bkg was more accurate than zap-bkg (Fig. 5). In the CLS model, the bias of the calculated $N_2O$ concentration using syn-bkg was significantly lower than zap-bkg based on the same spectral window ($W_N1$-3; P-values < 0.05) (Fig. 5). For $N_2O$ quantification, we only used the P-branch of $N_2O$ (2130-2223.7 $cm^{-1}$) to calculate $N_2O$ concentrations because the R-branch feature of $N_2O$ (2224-2280 $m^{-1}$) was strongly overlapped by the feature of $CO_2$ (2224-2450 $cm^{-1}$) in field spectra. Syn-bkg

better simulated the appropriate SB background over P- and R-branches of $N_2O$ than zap-bkg which simply removed the P-branch of $N_2O$. Thus, syn-bkg can generate the $N_2O$ absorbance (P-branch) without losing $N_2O$ absorbance intensity (Fig. 3d) as well as the accuracy of the calculated $N_2O$.

The syn-bkg method, and the integrated window of 2215.8-2223.7 $cm^{-1}$ and 2188.7-2204.1 $cm^{-1}$ ($W_N3$) were considered as

the optimal combination for $N_2O$ quantifications using the CLS models (i.e., lowest bias = -5.6$\pm$2.6 % in CLS, Fig.5b). This optimal combination was also used in the PLS model to predict $N_2O$ concentrations. The mean bias of the calculated $N_2O$ reduced from -5.6 % (CLS model) to -0.3 % (PLS model) (Fig. 5c). As compared to the CLS model, the PLS model significantly improved the accuracy of the calculated $N_2O$ (P < 0.05) because the PLS algorithm can extract useful latent factors from the $N_2O/H_2O$ mixing spectra (e.g. the contribution of water vapour to $N_2O$). The PLS model, however, led to

higher variability in the calculated $N_2O$ than the CLS model based on the same window (Fig. 5c), indicating that the PLS model was more sensitive to the changed environment than the CLS model.



### 3.3 Carbon dioxide (400 ppmv)

For $CO_2$ estimations, three spectral windows were used in 2070-2085 $cm^{-1}$ (Fig. 4c). The accuracy of the calculated $CO_2$ concentrations was also improved by narrowing the spectral window (Fig. 6). In the CLS model, the broadest window ($W_C1$: 2070-2084 $cm^{-1}$) led to an underestimate of 6.4(±4.1) % in $CO_2$ calculations using zap-bkg. This bias reduced by narrowing the window to $W_C2$ (2075.5-2084 $cm^{-1}$) (i.e. 0.1±4.2 % underestimate). The calculated bias of $CO_2$ concentrations was -4.7(±2.4) % using $W_C1$ and syn-bkg. This bias can be reduced to -0.3(±2.4) % using $W_C2$. The most confined window ($W_C3$: 2075.5-2080.5 $cm^{-1}$) resulted in greater biases than $W_C2$, and particularly in conjunction with zap-bkg (i.e. 3.2±3.4 % bias) (Fig. 6). Thus, the range from 2075.5 to 2084 $cm^{-1}$ ($W_C2$) was considered as the optimal window for $CO_2$ estimations (Fig. 4).

Zap-bkg conjoined with the optimal window (i.e. $W_N3$ or $W_C2$) in the CLS model led to greater underestimates in $N_2O$ than $CO_2$ calculations (Bias: -10±2.3%, Fig.5a vs. -0.1±4.2%, Fig.6a). Since the absorbance features of $CO_2$ at 2076.9 $cm^{-1}$ (the band center) was less complicated than the P-branch of $N_2O$ from 2170 to 2224 $cm^{-1}$, the $CO_2$ absorbance converted by zap-bkg was similar with syn-bkg (Fig. 3c and 3d). Therefore, the calculated bias (Fig. 6) showed that there was no significant difference between zap- and syn-bkg methods for $CO_2$ concentration calculations. Generally speaking, zap-bkg showed a similar trend with syn-bkg by narrowing the spectral window. Zap-bkg led to the higher variability in the calculated $CO_2$, indicating that simply removing the $CO_2$ feature by the linear function potentially resulted in biases for $CO_2$ quantification.

The other potential region for $CO_2$ quantification was within 722-800 $cm^{-1}$ (the R-branch of $CO_2$ $v_2$ band, Fig. 3a). Various windows were examined to calculate $CO_2$ concentration using the CLS model in this region, and the $CO_2$ concentrations were 40-70 % underestimated no matter which window was used in conjunction with zap-bkg. The minimum calculated mean bias was -9.0(±2.9) % by incorporating two windows of 723-727.7 $cm^{-1}$ and 732-738.5 $cm^{-1}$ in conjunction with syn-bkg (data not shown). As compared with the results from 2070-2084 $cm^{-1}$ (Fig. 4c), 722-800 $cm^{-1}$ window resulted in a significant underestimations in $CO_2$ calculations because 1) more water vapour features interfered with the R-branch of $CO_2$ features in 722-800 $cm^{-1}$ than $CO_2$ in 2070-2084 $cm^{-1}$, and 2) it was difficult to simulate the appropriate background at the low wavenumber region in the SB spectrum.

### 3.4 Diurnal $N_2O/CO_2$ estimations

The optimal quantitative approach of leading to the minimum bias in $N_2O$ estimations was to use syn-bkg and $W_N3$ window in the PLS model (Fig. 5c); the optimal approach for $CO_2$ estimations was to use the syn-bkg procedure and $W_C2$ in the CLS model (Fig. 6b). These optimized methods were used to estimate the $N_2O$ and $CO_2$ concentrations from the OP-FTIR spectra collected from 9th to 19th in June 2014 (Fig. 7). The diurnal fluctuation in $N_2O$ and $CO_2$ concentrations were corresponding to the diurnal changes of the wind speed and air temperature. The higher $N_2O/CO_2$ concentrations were usually measured during the night because of $N_2O$ and $CO_2$ accumulations. The accumulation of $N_2O/CO_2$ occurred near the ground when





turbulent mixing was low, resulting from the decreasing buoyancy from the ground surface (i.e. a stable atmosphere). The greater density of air parcel due to the decreasing temperature also led to the gas accumulation. The diurnal variation in $CO_2$ was greater than $N_2O$ (Fig. 7b), and we hypothesized because of the multiple sources of $CO_2$. $N_2O$ was mostly produced from soils via microbial nitrification and denitrification, but $CO_2$ was emitted via soil respiration (including microbes and

corn root) as well as the respiration from grass and corn leaves.

Mixing of the surface layer of the air tended to result in greater homogeneity along the optical path. Under low wind speed, the presumably poorly-mixed air increased the variability of the path-averaged $N_2O/CO_2$ concentrations along the optical path, resulting in the difference between the 50 m S-OPS and the 150 m OP-FTIR. The calculated biases of $N_2O$ and $CO_2$

were 1.3(±2.6) % (n=363) and -0.7(±6.0) % (n=327), respectively, while the mean wind velocity ranged from 0.1 to 8.4 m·s$^{-1}$ (Fig. 7). The variability of the calculated biases of $N_2O$ and $CO_2$ were reduced when the data that were collected under the low wind speed (<1.7 m·s$^{-1}$) were excluded, i.e. $bias_{N_2O}$=1.4±2.3 % (n=295) and $bias_{CO_2}$=-1.0±2.8 % (n=269).

## 4 Conclusion

We have developed various methods to quantify concentrations of nitrous oxide and carbon dioxide using open-path FTIR

based on combinations of single beam backgrounds (i.e. zap-bkg and syn-bkg), analytical windows, and chemometric multivariate calibration models (i.e. CLS and PLS). It is challenging to generate the P-branch $N_2O$ absorbance within 2170-2223.7 cm$^{-1}$ to predict $N_2O$ accurately but feasible to generate $CO_2$ absorbance in 2075.5-2084 cm$^{-1}$ for $CO_2$ prediction using the zap-bkg method. The principle for selecting spectral window is to use the region with less water vapour features, yet over confining the analytical region may lead to biases in gas predictions. The CLS model, the most common approach used for

gas retrievals in OP-FTIR commercial packages, underestimates $N_2O$ concentrations but can predict $CO_2$ accurately within the absolute humidity of 5,000-20,000 ppmv and the temperature of 10-35 ℃. In this study, the optimal method for $N_2O$ quantification is to use the combination of syn-bkg, two bands window (2188.7-2204.1 + 2215.8-2223.7 cm$^{-1}$), and the PLS model ($N_2O$ bias = 1.4±2.3 %). The optimal method for $CO_2$ quantification is to use the combination of syn-bkg, 2075.5-2084 cm$^{-1}$ window, and the CLS model ($CO_2$ bias = -1.0±2.8 %). We provide comprehensive methods of $N_2O/CO_2$ analyses

for the increasing OP-FTIR users who are interested in greenhouse gas emissions from agricultural fields.

*Acknowledgments.* This study was supported by the United States Department of Agriculture National Institute for Food and Agriculture Grant No. 13-68002-20421, Indiana Corn Marketing Council Grant No. 12076053, Purdue University Climate Change Research Center. We would like to thank Dr. Tony Vyn and Terry West for the crop and field management, and

Austin Pearson for data collection and analyses.




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

**Figure 1.** Schematic of the instrumentation used to assess the accuracy of $N_2O$ and $CO_2$ concentration determined by OP-FTIR in this study. DFG $N_2O$ and LI-840 $CO_2$ analyzers combined with the synthetic open path air-sampling system (S-OPS) were used to measure the
15   'actual' path-averaged $N_2O/CO_2$ concentrations and benchmark the $N_2O$ and $CO_2$ concentrations calculated from OP-FTIR spectral analyses. The humidity, air temperature, and wind information were measured from the weather station.

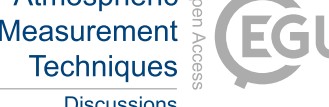

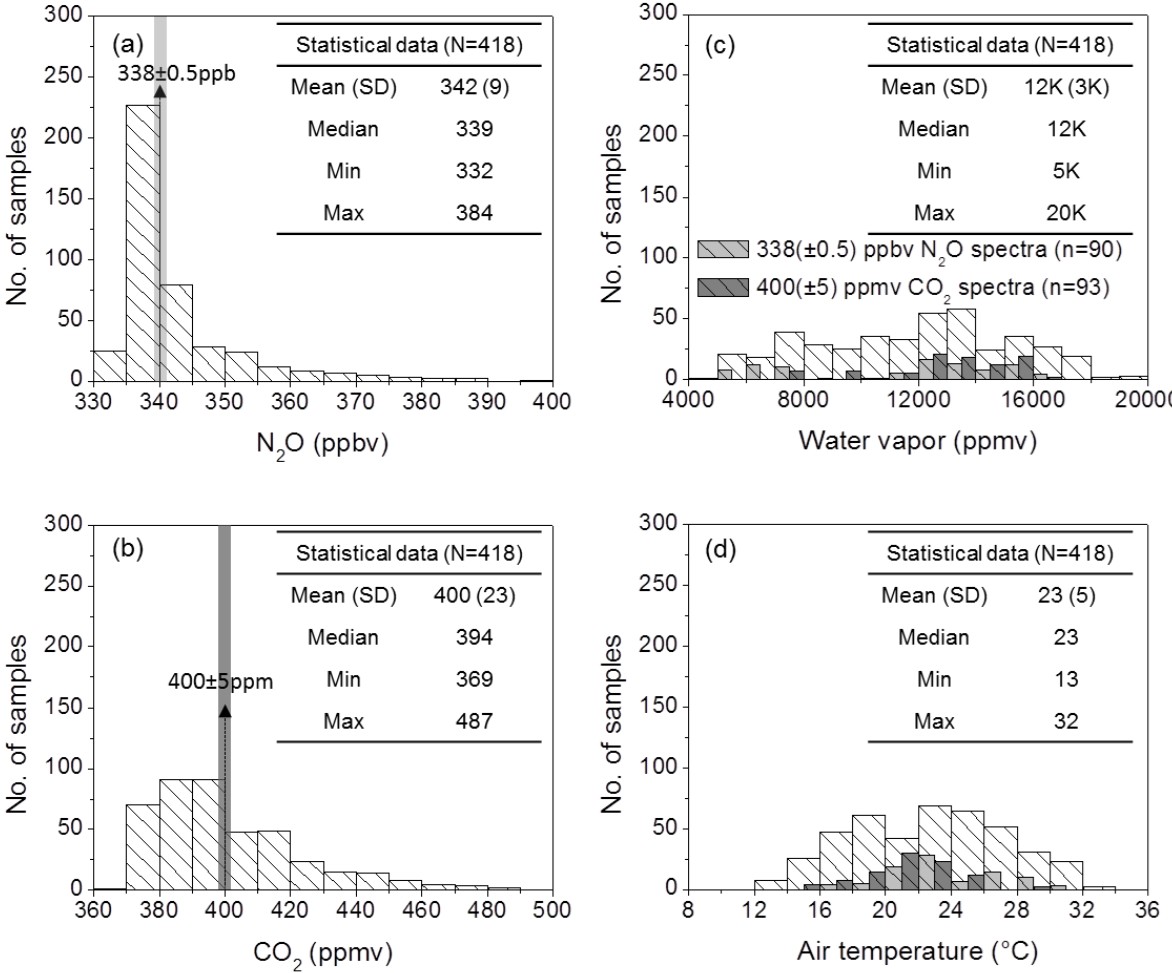

**Figure 2.** The 30-min averaged concentrations of (a) $N_2O$ and (b) $CO_2$ were measured using $N_2O$ and $CO_2$ analyzers by sampling the air from S-OPS, and the 30-min averages of (c) water vapour content and (d) air temperature were also measured from the on-site weather station during 9[th]-19[th] in June, 2014. The concentrations of $N_2O$, $CO_2$, and water vapour showed in these figures were measured while the air was well-mixing (U > 1.7 m·s[-1]). The light gray bars mean the OP-FTIR spectra containing 338($\pm$0.5) ppbv $N_2O$ and the dark gray barks mean the OP-FTIR spectra containing 400($\pm$5) ppmv $CO_2$. Both the selected spectra ($N_2O$ 338 ppbv, n=90; $CO_2$ 400 ppmv, n=93) covered the broad ranges of water vapour and air temperature and were used to assess the sensitivity of the OP-FTIR quantitative methods to the dynamic ambient factors.





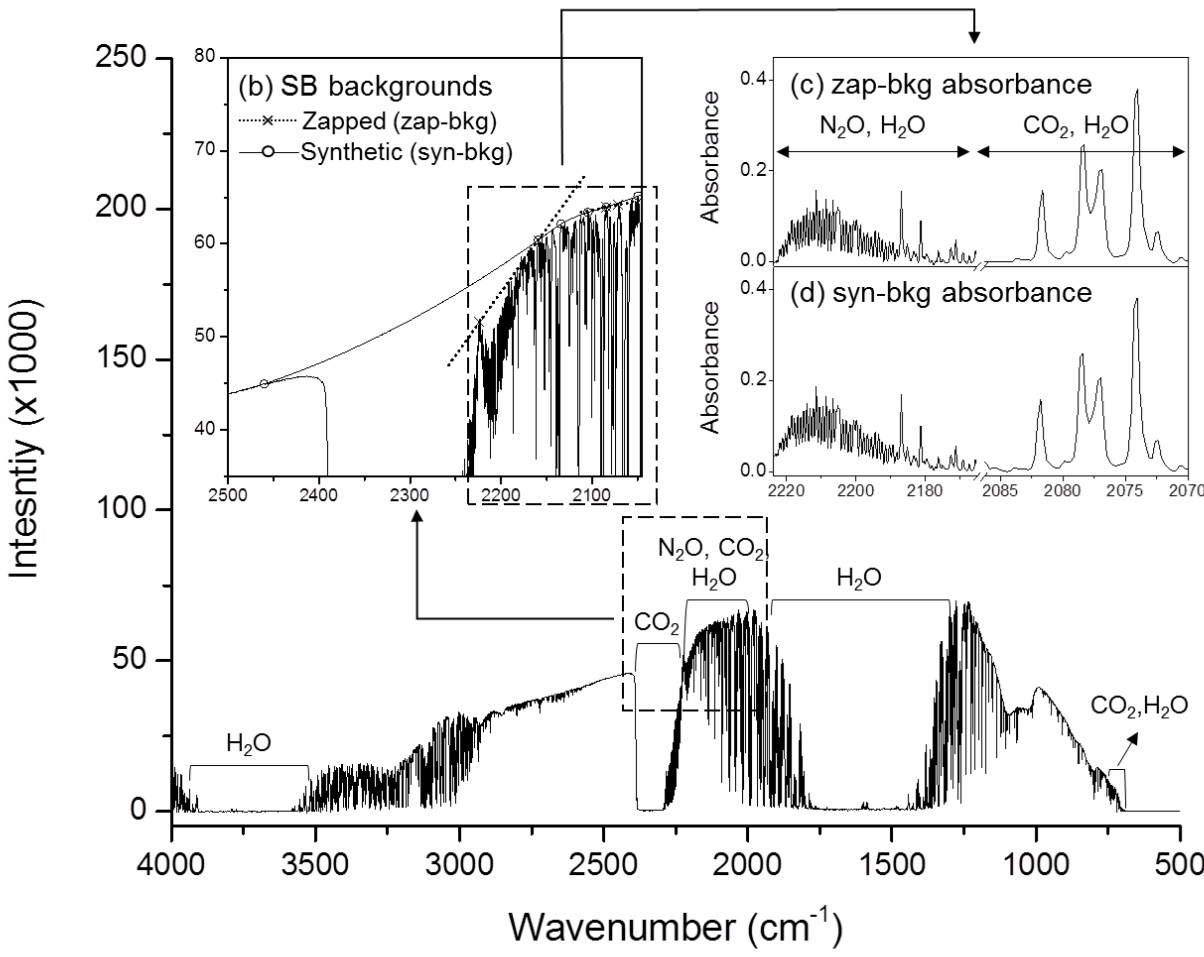

**Figure 3.** The illustrations of (a) a field single beam (SB) OP-FTIR spectrum containing the regions of $N_2O$, $CO_2$, and water vapour was collected through an optical path length of 300 m; (b) a zapped and a synthetic SB backgrounds (zap-bkg and syn-bkg) were generated from this field SB spectrum and used to convert the sampled SB spectrum to (c) the absorbance spectra that allow us to calculate $N_2O/CO_2$ concentrations using Beer-Lambert law.





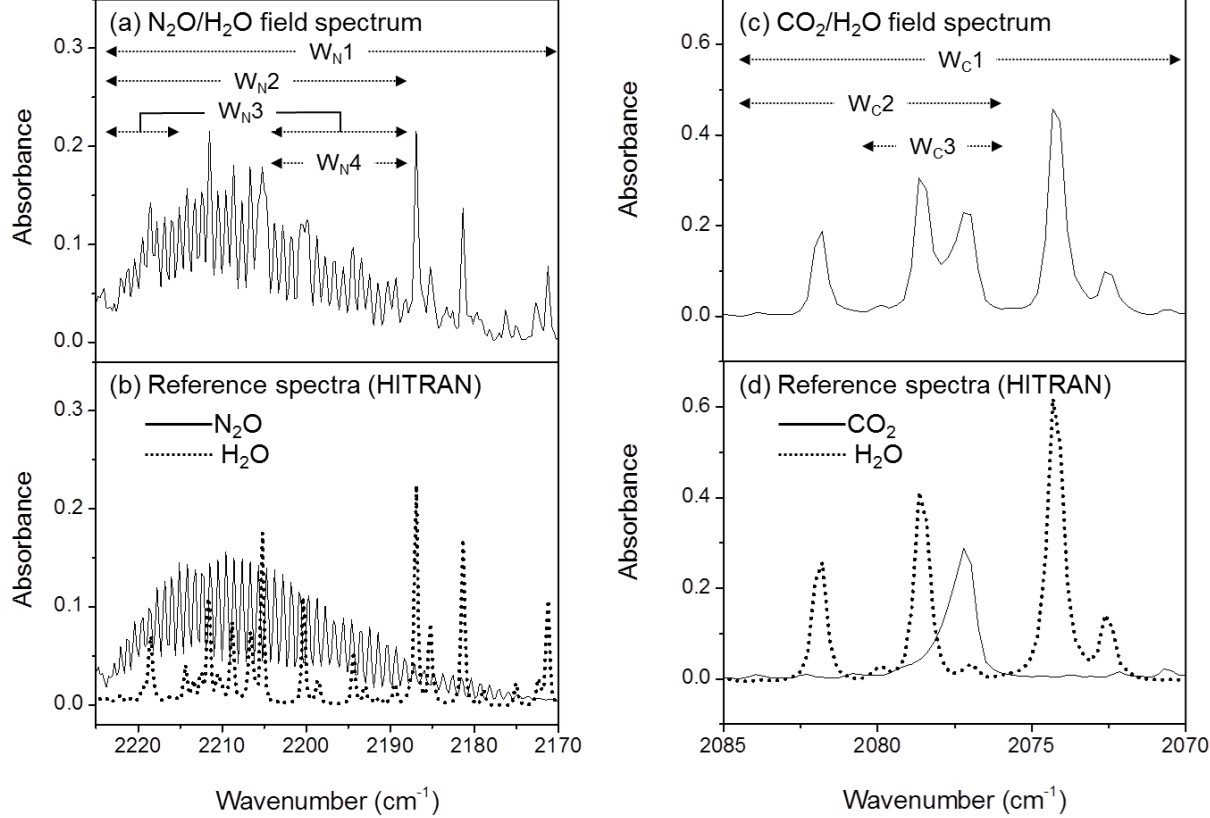

**Figure 4.** Field and HITRAN reference absorbance spectra: (a) Field spectrum containing the features of $N_2O$ and water vapour, (b) reference spectra of $N_2O$ and water vapour at 2170 - 2225 cm$^{-1}$, (c) field spectrum containing the features of $CO_2$ and water vapour, and (d) reference spectra of $CO_2$ and water vapour at 2070 - 2085 cm$^{-1}$. $W_N(1-4)$ and $W_C(1-3)$ denote the spectral windows used to calculate $N_2O$ and $CO_2$ concentrations from field spectra.





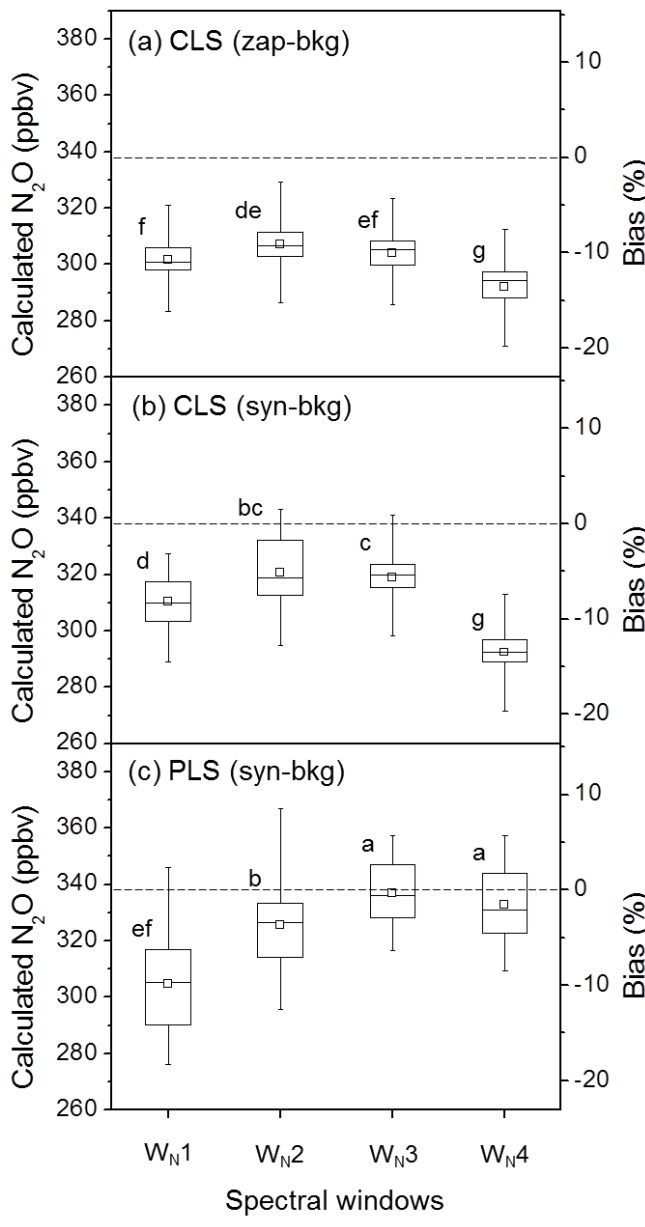

**Figure 5.** The box plots of the calculated $N_2O$ concentrations and the corresponding biases from a series of OP-FTIR spectra (n=90) that contain 338 ppbv $N_2O$ with the changed humidity and air temperature using different SB background-processing methods (i.e. zap-bkg and syn-bkg), and four spectral windows ($W_N$1-4) in the CLS and PLS models: (a) zap-bkg + CLS model, (b) syn-bkg + CLS model, and (c) syn-bkg + PLS model. The plot displays the mean (□), median (—), interquartile ranges (box), and extreme values (whiskers). Different letters indicate the significant difference (P < 0.05) among the means calculated by different quantitative methods by the least significant difference (LSD).



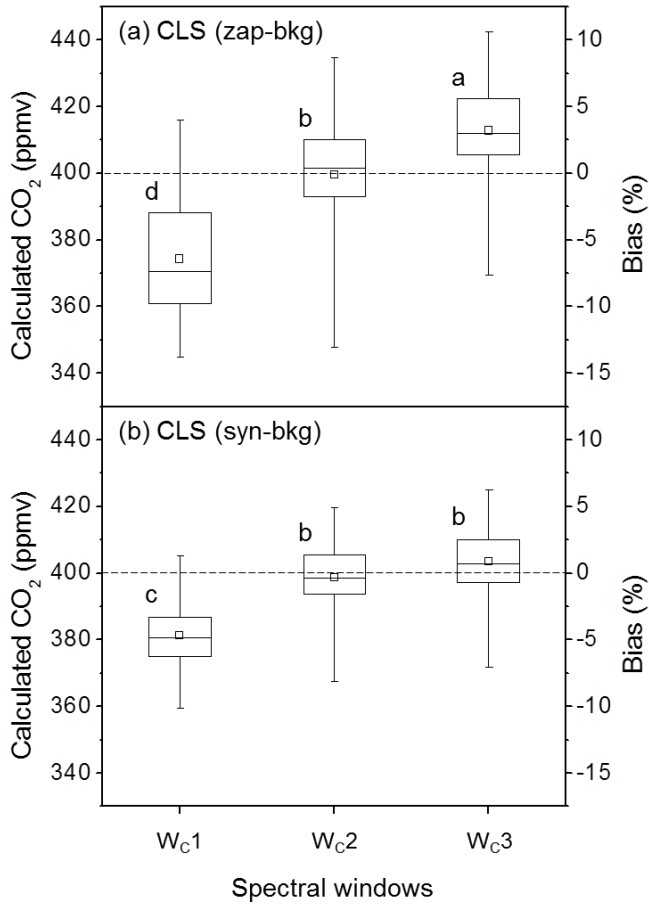

**Figure 6.** The box plots of the calculated $CO_2$ concentrations and the corresponding biases from a series of OP-FTIR spectra (n=93) that contain 400 ppmv $CO_2$ with the changed humidity and air temperature using different SB background-processing methods (i.e. zap-bkg and syn-bkg), and three spectral windows ($W_C$1-3) in the CLS model: (a) zap-bkg, and (b) syn-bkg. The plot displays the mean (□), median (—), interquartile ranges (box), and extreme values (whiskers). Different letters indicate the significant difference (P < 0.05) among the means calculated by different quantitative methods by the least significant difference (LSD).



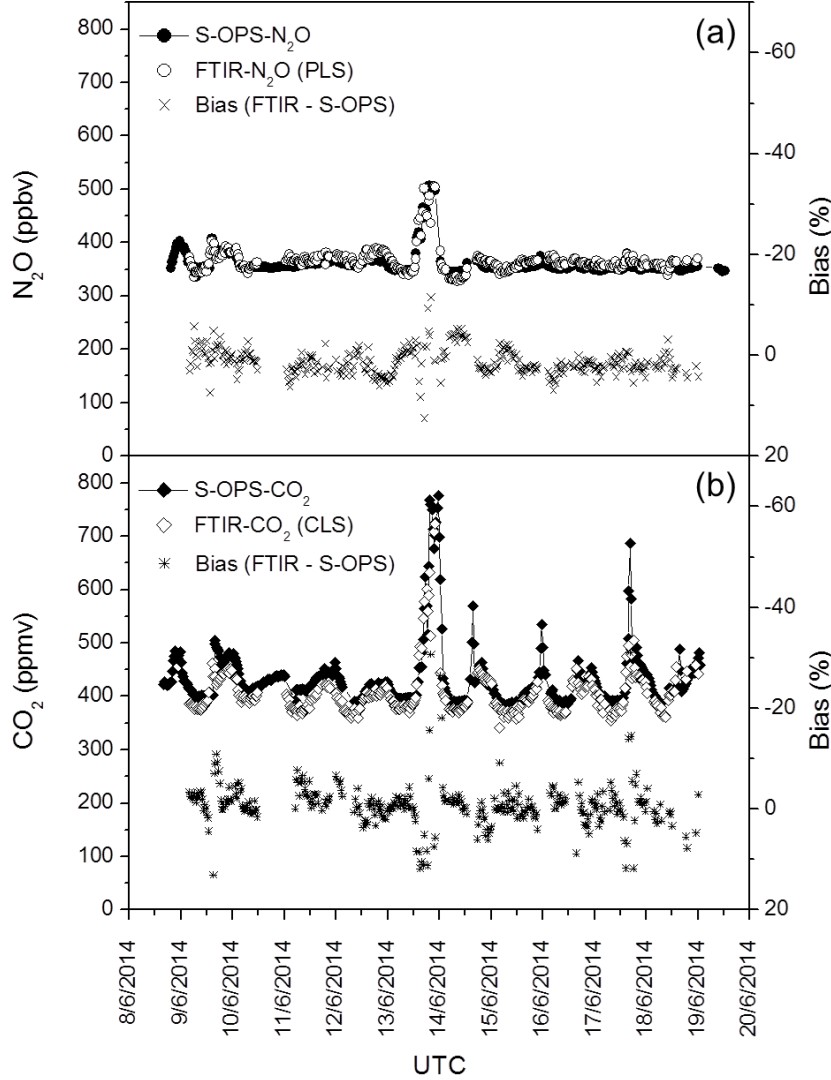

**Figure 7.** $N_2O$ and $CO_2$ concentrations from 9[th] to 19[th] in June 2014: (a) $N_2O$ concentrations measured from S-OPS using the DFG $N_2O$ analyzer and calculated from OP-FTIR using the optimal methods (syn-bkg + $W_N3$ + the PLS model), and the corresponding biases, and (b) $CO_2$ concentrations measured from S-OPS using LI-840 $CO_2$ analyzer and calculated from OP-FTIR using the optimal method (syn-bkg + $W_C2$ + the CLS model), and the corresponding bias.