# Peer review of "Application of Open Path Fourier Transform Infrared Spectroscopy (OP-FTIR) to Measure Greenhouse Gas Concentrations from Agricultural Soils"

_Atmospheric Measurement Techniques, 2018_

## Author Comment (AC1) · 13 Dec 2018

The mixing ratios of N2O (ppbv) and CO2 (ppmv) that were determined by S-OPS and calculated from OP-FTIR spectra were shown in the Figure-7. The measured and calculated mixing ratios needed to be corrected by the humidity content in the air (dry air correction). The original Figure-7 showed the dry-air corrected concentrations of N2O (both from S-OPS and FTIR) and CO2 from S-OPS, but CO2 concentrations that were calculated from OP-FTIR were not corrected by humidity content by accident. We updated the Figure-7 (attached) with the dry-air corrected CO2 concentrations that

were calculated from FTIR spectra (FTIR-CO2 CLS).

For the caption of the Figure-2, the air was considered well-mixed when the mean wind velocity was above 1.5 m/s instead of 1.7 m/s.
* * *
[Figure]

**Fig. 1.** The updated Figure-7: N2O and CO2 concentrations from 9th to 19th in June 2014.

---

## Referee Comment (RC1) · Anonymous Referee #2 · 27 Dec 2018

This is a good investigation, and it has scientific and practical benefits to researchers in atmospheric monitoring. Before acceptance, there are several issues to be addressed.

In section 2.3, the authors stated that "we selected ninety spectra containing $338\pm0.5$ ppbv N2O and ninety-three spectra containing $400\pm5$ ppmv CO2 which were measured from the S-OPS. These group of spectra covered broad ranges of water vapour content and air temperature. N2O and CO2 concentrations were calculated from these selected spectra using different quantitative methods." What is the exact meaning of "ninety spectra containing $338\pm0.5$ ppbv N2O"? The purpose of the selection seems

to be "avoid the non-linear response of absorbance to the wide range of gas concentrations", then how could "338±0.5 ppbv N2O" serve this purpose?

In section 2.4, the authors stated that "Each sampled spectrum was acquired by co-adding 64 single-sided interferograms (IFGs) at a nominal resolution of 0.5 cm-1, which accounted for 32,000 data points were collected with the interval of 0.241 wavenumbers between data points, ..." For interferograms, the unit of the interval of data points is cm, not wavenumber.

In section 2.4, the authors employed some criteria to remove low-quality IFGs, which includes those of very intense centerburst. It is true that intense-centerburst IFGs result in severe non-linear response of MCT detectors. However, such IFGs have high signal-to-noise ratio, and would be valid once the non-linear detector response is corrected. The authors might be interested in a correcting method (L. Shao, P. R. Griffiths, Anal. Chem., 2008, 80(13), 5219)

In section 2.5.1, the authors stated using "a high-order fitting function" as the synthetic background. It is better to be specific about the function, is it a quadratic, cubic polynomial, or something else?

In section 2.5.2, some useful information about PLS models is not provided, such as the number of calibration spectra (to build the model), the concentration range that the model covers, the number of factors for the model.

In section 2.6, it is better to be specific about the statistical tests, are they t-tests or paired t-tests?

In section 3.2, the authors present the results of CLS (zap-bkg) and CLS (syn-bkg), and the result of PLS (syn-bkg). Why is the result of PLS (zap-bkg) absent?

It seems that the authors did not apply PLS to estimate the concentrations of CO2, as they did in case of N2O. The reason should be explained.

Fig. 7(b) is strange. As stated in the fig. 7(b), bias = FTIR - S-OPS. According to

this formula, the bias between 11/6/2014 and 12/6/2014 is negative, since the FTIR concentrations are clearly lower than S-OPS. But in the figure the corresponding bias is positive.

---

## Author Comment (AC2) · 15 Jan 2019

Thank you for your comments and suggestions. Our responses to the "Several issues should be addressed" were as follows:

1. In section 2.3, the authors stated that "we selected ninety spectra containing  $338\pm0.5$  ppbv N2O and ninety-three spectra containing  $400\pm5$  ppmv CO2 which were measured from the S-OPS. These group of spectra covered broad ranges of water vapour content and air temperature. N2O and CO2 concentrations were calculated

from these selected spectra using different quantitative methods." What is the exact meaning of "ninety spectra containing  $338\pm0.5$  ppbv N2O"? The purpose of the selection seems to be "avoid non-linear response of absorbance to the wide range of gas concentrations", then how could " $338\pm0.5$  ppbv N2O"serve this purpose?

Response: 1) The atmospheric N2O concentrations were simultaneously measured from the S-OPS and OP-FTIR along the same path. The S-OPS-measured N2O was used as a benchmark to examine the performance of the OP-FTIR quantitative methods. Thus, each spectrum was corresponding to a particular N2O concentration measured from the S-OPS. These quantitative methods were tested using the selected ninety OP-FTIR spectra where the S-OPS-measured N2O concentrations ranged from 337.4 – 338.5 ppbv. Likewise, ninety-three spectra where the CO2 concentrations ranged from 405.4 – 394.9 ppmv were selected. The statistics of N2O and CO2 concentrations were shown in the following table 1. 2) A broad range of the path-integrated concentration tends to result in a non-linear response of absorbance to concentration. The selected spectra contained particular N2O/CO2 concentrations but various water vapor contents and temperature. The non-linear response of absorbance to the changed water vapor content cannot be solved, but the effect of the changed N2O/CO2 on the non-linearity of absorbance can be minimized by constraining gas concentrations.

2. In section 2.4, the authors stated that "Each sampled spectrum was acquired by coadding 64 single-sided interferograms (IFGs) at a nominal resolution of 0.5 cm-1, which accounted for 32,000 data points were collected with the interval of 0.241 wavenumbers between data points, ..." For interferograms, the unit of the interval of data points is cm, not wavenumber.

Response: This sentence might be confused. It means that a resolution of 0.5 cm-1 accounting for a data point every 0.241 wavenumbers was used for acquiring SB spectra (400 – 4000 cm-1), and approximate 32,000 data points were in the interferogram.
3. In section 2.4, the authors employed some criteria to remove low-quality IFGs, which includes those of very intense centerburst. It is true that intense-centerburst IFGs result in severe non-linear response of MCT detectors. However, such IFGs have high signal-to-noise ratio, and would be valid once the non-linear detector response is corrected. The authors might be interested in correcting method (L. Shao, P.R. Griffiths, Anal. Chem., 2008, 80(13), 5219).

Response: The maximum A/DC capacity in this study was approximately 2.49 Volts. The optical path length of the OP-FTIR was 300 meters. The maximum or minimum value of the IFG centerburst in this study located between 0.61-1.14 volts, which prevented the MCT detector from saturation as well as avoided the non-linear response of the detector.

4. In section 2.5.1, the authors stated using "a high-order fitting function" as the synthetic background. It is better to be specific about the function, is it a quadratic, cubic polynomial, or something else?

Response: Numerous data points were selected from the field SB spectrum. A polynomial function was used to fit the field spectrum to synthesize the SB background without features of gas absorption.

5. In section 2.5.2, some useful information about PLS models is not provided, such as the number of calibration spectra (to build the model), the concentration range that the model covers, the number of factors for the model.

Response: Sixty mixed-gas (i.e., N2O + water vapor) spectra were collected from the lab-based FTIR joined with the multi-pass gas cell (the optical path length of 33 meters). Concentrations of N2O and water vapor ranged from 0.3 - 0.7 ppmv and 7000 – 30,000 ppmv, respectively. More details of the calibration spectra were shown in the following Table 2.

6. In section 2.6, it is better to be specific about the statistical tests, are they t-test or
**paired t-test?**

Response: For N2O analysis, twelve quantitative models that were used to calculate N2O concentration from ninety OP-FTIR spectra were examined to optimize the combinations of SB backgrounds (i.e., zap- and syn-bkg), multivariate models (i.e., CLS and PLS), and analytical windows (i.e., WN1-WN4). In order to compare the means of the twelve populations, the Fisher's Least significant difference (LSD) was used for multiple comparisons ( $\alpha = 0.05$ ). Likewise, the LSD was also used to compare six population means for the CO2 analysis.

7. In section 3.2, the authors present the result of CLS (zap-bkg) and CLS (syn-bkg), and the result of PLS (syn-bkg). Why is the result of PLS (zap-bkg) absent? It seems that the authors did not apply PLS to estimate the concentrations of CO2, as they did in case of N2O. The reason should be explained.

Response: 1) The syn-bkg is one of the recommended methods for converting the SB to absorbance spectra, but the zap-bkg was the new proposed method. Thus, the syn-bkg was used to examine the feasibility as well as the performance of the zap-bkg. The identical field SB spectra, analytical windows, and CLS model were used to calculate gas concentrations from the zap- and syn-bkg converted absorbance spectra. For both N2O and CO2 analyses, the zap-bkg method led to higher biases in concentration calculations than the syn-bkg using CLS models. In section 3.2, the zap-bkg resulted in great underestimations (i.e., Bias > 9%) for N2O quantification and the syn-bkg improved the quantitative accuracy. Applying the PLS to quantify gas concentration from the zap-bkg converted spectra unlikely improve the quantitative accuracy. For simplification, we did not report the results of the integrated uses of the zap-bkg and PLS model. 2) Compared with N2O analysis, the integration of the synbkg and CLS model provided decent predictions for CO2 concentrations, which was presumably due to the simplicity of the CO2 absorption feature at 2170-2085 cm-1. This combination, however, did not provide the same accuracy for N2O predictions. Therefore, we only applied the PLS model for N2O predictions, and this model did
improve its accuracy.

8. Fig.7(b) is strange. As stated in the Fig.7(b), bias = FTIR – S-OPS. According to this formula, the bias between 11/6/2014 and 12/6/2014 is negative, since the FTIR concentrations are clearly lower than S-OPS. But in the figure the corresponding bias is positive.

Response: The Y-axis of bias (%) is reverse, so the biases should be negative. Also, the updated Fig.7 and the explanation for updating Fig7(b) were described in the author comment (AC1).
| Casas                   | Statistics of the S-OPS measured concentrations |     |       |        |       |    |  |  |
|-------------------------|-------------------------------------------------|-----|-------|--------|-------|----|--|--|
| Gases                   | Mean                                            | SD  | Max   | Median | Min   | n  |  |  |
| N 2 O (ppbv) | 337.9                                           | 0.3 | 338.5 | 337.9  | 337.4 | 90 |  |  |
| $CO_2$ (ppmv)           | 399.8                                           | 3.0 | 405.4 | 398.9  | 394.9 | 93 |  |  |

**Fig. 1.** Table 1. The S-OPS measured concentrations of N2O/CO2 used for OP-FTIR quantitative method evaluations.

AMTD

Interactive

(a) N2O/water vapor concentrations

| Sportratt | N 2 O | N 2 O | Water vapor | Water vapor | Sportratt | N 2 O | N 2 O | Water vapor | Water vapor |
|-----------|------------------|------------------|-------------|-------------|-----------|------------------|------------------|-------------|-------------|
| Spectra#  | (ppm)            | (ppm*m)          | (ppm)       | (ppm*m)     | spectra#  | (ppm)            | (ppm*m)          | (ppm)       | (ppm*m)     |
| 1         | 0.31             | 10.23            | 7330.60     | 241909.89   | 31        | 0.31             | 10.23            | 20720.00    | 683760.00   |
| 2         | 0.31             | 10.23            | 7334.38     | 242034.64   | 32        | 0.31             | 10.23            | 20720.00    | 683760.00   |
| 3         | 0.31             | 10.23            | 7344.42     | 242365.75   | 33        | 0.31             | 10.23            | 20720.00    | 683760.00   |
| 4         | 0.40             | 13.20            | 7345.97     | 242417.03   | 34        | 0.40             | 13.20            | 21166.00    | 698478.00   |
| 5         | 0.40             | 13.20            | 7335.69     | 242077.81   | 35        | 0.40             | 13.20            | 21166.00    | 698478.00   |
| 6         | 0.40             | 13.20            | 7384.26     | 243680.45   | 36        | 0.40             | 13.20            | 21166.00    | 698478.00   |
| 7         | 0.50             | 16.50            | 7425.25     | 245033.38   | 37        | 0.50             | 16.50            | 21352.00    | 704616.00   |
| 8         | 0.50             | 16.50            | 7435.31     | 245365.38   | 38        | 0.50             | 16.50            | 21352.00    | 704616.00   |
| 9         | 0.50             | 16.50            | 7428.77     | 245149.42   | 39        | 0.50             | 16.50            | 21352.00    | 704616.00   |
| 10        | 0.60             | 19.80            | 7472.43     | 246590.20   | 40        | 0.60             | 19.80            | 22409.00    | 739497.00   |
| 11        | 0.60             | 19.80            | 7561.33     | 249524.05   | 41        | 0.60             | 19.80            | 22409.00    | 739497.00   |
| 12        | 0.60             | 19.80            | 7561.16     | 249518.13   | 42        | 0.60             | 19.80            | 22409.00    | 739497.00   |
| 13        | 0.70             | 23.10            | 7428.18     | 245129.78   | 43        | 0.70             | 23.10            | 25814.00    | 851862.00   |
| 14        | 0.70             | 23.10            | 7387.42     | 243784.78   | 44        | 0.70             | 23.10            | 25814.00    | 851862.00   |
| 15        | 0.70             | 23.10            | 7359.32     | 242857.45   | 45        | 0.70             | 23.10            | 25814.00    | 851862.00   |
| 16        | 0.31             | 10.23            | 15367.62    | 507131.41   | 46        | 0.31             | 10.23            | 26786.16    | 883943.19   |
| 17        | 0.31             | 10.23            | 15372.82    | 507303.06   | 47        | 0.31             | 10.23            | 26584.12    | 877275.94   |
| 18        | 0.31             | 10.23            | 15360.80    | 506906.44   | 48        | 0.31             | 10.23            | 26597.24    | 877708.81   |
| 19        | 0.40             | 13.20            | 15595.24    | 514643.03   | 49        | 0.40             | 13.20            | 30446.93    | 1004748.69  |
| 20        | 0.40             | 13.20            | 15704.48    | 518247.72   | 50        | 0.40             | 13.20            | 30738.75    | 1014378.88  |
| 21        | 0.40             | 13.20            | 15708.70    | 518387.22   | 51        | 0.40             | 13.20            | 30386.46    | 1002753.19  |
| 22        | 0.50             | 16.50            | 15521.94    | 512224.06   | 52        | 0.50             | 16.50            | 29310.16    | 967235.44   |
| 23        | 0.50             | 16.50            | 15678.90    | 517403.75   | 53        | 0.50             | 16.50            | 28955.07    | 955517.25   |
| 24        | 0.50             | 16.50            | 15771.27    | 520452.06   | 54        | 0.50             | 16.50            | 28851.81    | 952109.88   |
| 25        | 0.60             | 19.80            | 15766.92    | 520308.47   | 55        | 0.60             | 19.80            | 28499.03    | 940467.94   |
| 26        | 0.60             | 19.80            | 15707.03    | 518332.09   | 56        | 0.60             | 19.80            | 28247.49    | 932167.31   |
| 27        | 0.60             | 19.80            | 15859.43    | 523361.28   | 57        | 0.60             | 19.80            | 27876.50    | 919924.50   |
| 28        | 0.70             | 23.10            | 16033.97    | 529121.00   | 58        | 0.70             | 23.10            | 28584.89    | 943301.38   |
| 29        | 0.70             | 23.10            | 15967.60    | 526930.88   | 59        | 0.70             | 23.10            | 29724.95    | 980923.31   |
| 30        | 0.70             | 23.10            | 15887.32    | 524281.53   | 60        | 0.70             | 23.10            | 29897.34    | 986612.19   |

(b) The number of factors in PLS models

|                                                     | No. of factors in PLS |             |  |  |
|-----------------------------------------------------|-----------------------|-------------|--|--|
| Analytical window (cm -1 )               | $N_2O$                | water vapor |  |  |
| W N 1: 2170.0 - 2223.7                   | 3                     | 5           |  |  |
| W N 2: 2188.5 - 2223.7                   | 4                     | 4           |  |  |
| W N 3: 2188.5 - 2204.1 + 2215.8 - 2223.7 | 4                     | 4           |  |  |
| W N 4: 2188.5 - 2204.1                   | 5                     | 4           |  |  |

**Fig. 2.** Table 2. Sixty mixed-gases calibration spectra were used to build PLS models for N2O quantification: (a) concentrations of N2O and water vapor, and (b) the number of factors used in PLS models.

comment

---

## Referee Comment (RC2) · Anonymous Referee #3 · 25 Mar 2019

I really appreciate the here presented application of OP-FTIR for observation of characteristic GHG emissions from soils and agricultural landscapes. To my knowledge there is only a minor amount of papers describing the usage of this remote sensing technology in this area of interest and I fully agree with the authors concerning the potential of this method to remotely detect the emission volatile gas components in a fast and effective way.

From my side I would like to mention a few minor comments:

[Figure]

- Title: "...GHG concentrations from agricultural soils" - you measure GHG concentrations in ambient air above ground surface - these concentrations can stem from emissions from the soil (including roots and microorganisms) or from the above ground vegetation. You do not measure fluxes - so be careful within your introduction - to estimate fluxes the concentration is only one of the variables needed!!!

- Page 2, line 10 ff: The authors mention chamber measurements as the most common way to investigate emissions from soils. In the same time they point out the relatively small footprint as the main limit of this method. However, I would like to introduce in this context the opportunity to measure GHG fluxes on larger scale using the Eddy Covariance flux measurements. This method is also an established method nowadays and a common micro-meteorological technique with an increased footprint to determine emissions for instance of $CO_2$, $CH_4$ and water from soils and vegetation. (e.g., Baldocchi, D. (2003): Assessing the eddy covariance technique for evaluating carbon dioxide exchange rates of ecosystems: past, present and future. https://doi.org/10.1046/j.1365-2486.2003.00629.x). Concerning to comment No. 1 - all methods are based on the measurement of the concentrations of GHGs and have their own processes to obtain emission rates.

- General comment: Time series on GHG concentrations at S-OPS including measured ambient air conditions (to have an idea about the variability of wind speed, direction, air temp, etc.) would be helpful.

- Page 4, line 24: How often did you acquire single beam spectra (how many spectra did you measure during the operational period - also to make the number of 877 valid OP-FTIR spectra more valuable)?

- Page 5, line 17: What do you mean here: "we selected ninety spectra containing 338 ppbv $N_2O$ and ninety-three spectra containing 400 ppmv $CO_2$?" These spectra do contain the same concentration like the measurements at the S-OPS? Is the impact of IR absorbance due to water vapor within this spectra not so significant (my interpretation of figure 2d)? (In Figure 2 the readers find the histograms of 418 half-hour average-intervals? In Fig.2a the light grey line is located at x = 340 ppb and not at x = 338 ppb, by the way ...)

- Page 6, line 23: I see the potential and limitation of the here discussed methods to obtain a target gas free background spectrum. In my opinion, using one background spectrum per day for the zap-bkg determination is to less due to the extensive impact of changing environmental conditions on the measured IR spectra (which is surely occurring during the day in ambient air conditions like air humidity, pressure variability, ....).

- Page 6, line 30: what is meant with: all data points are stored as one data file? You calculated from each SB spectra the synthetic background and store this in the same manner like the original data spectra and us these files for the calculation of abs-spectra? How many data points are used to determine the smooth background spectra function (which order of polynom)?

- Page 8, line 27: humidity

- Page 9 / 10: Figure 5 and 6 imply, that the authors did not show all the results. To evaluate the presented methods in order to agree with the proposed "optimal approaches", a comparison of all concentration estimations (CLS, PLS, used spectral windows and background spectra) should be shown (or at least mentioned for instance as a table). Otherwise, the assessment is very hard for the reader... (For instance in chapter 3.3 the PLS for $CO_2$ is missing.)

---

## Author Comment (AC3) · 22 Apr 2019

Dear referees, we truly appreciated that your valuable comments that help us clarify some concepts in this manuscript. My responses to your questions are as follows,

1. Title: "...GHG concentrations from agricultural soils" - you measure GHG concentrations in ambient air above ground surface - these concentrations can stem from emissions from the soil (including roots and microorganisms) or from the above ground vegetation. You do not measure fluxes - so be careful within your introduction - to

[Figure]

estimate fluxes the concentration is only one of the variables needed!!!

Response: I would modify this title to 'Application of Open Path Fourier Transform Infrared Spectroscopy (OP-FTIR) to Measure Greenhouse Gas Concentrations at a Maize Cropping System '

2. Page 2, line 10 ff: The authors mention chamber measurements as the most common way to investigate emissions from soils. In the same time they point out the relatively small footprint as the main limit of this method. However, I would like to introduce in this context the opportunity to measure GHG fluxes on larger scale using the Eddy Covariance flux measurements. This method is also an established method nowadays and a common micro-meteorological technique with an increased footprint to determine emissions for instance of CO2, CH4 and water from soils and vegetation. (e.g., Baldocchi, D. (2003): Assessing the eddy covariance technique for evaluating carbon dioxide exchange rates of ecosystems: past, present and future. https://doi.org/10.1046/j.1365- 2486.2003.00629.x). Concerning to comment No. 1 - all methods are based on the measurement of the concentrations of GHGs and have their own processes to obtain emission rates.

Response: It is a good idea to point out the eddy covariance and its advantage (e.g., larger footprint) for gas flux measurements. I would briefly introduce this method into the context before I submit the final version. Chamber measurements are the most common method for soil gas emission measurements because this method also provides numbers of strengths. One of the advantages is that chamber is sensitive enough to make comparisons of gas emissions in different treatments (e.g., field and N management practices in small plots (< 3 ha)), which is challenging to the eddy covariance. The OP-FTIR combined with inversion dispersion techniques (e.g., backward Lagrangian stochastic dispersion model) is capable of measuring gas emissions frequently and with a field-scale footprint (1-3 ha), that can both compensate the limitations of chamber measurements and measure gas emissions from different treatment plots. That is the reason why we did not introduce the eddy covariance in the first

place.

3. General comment: Time series on GHG concentrations at S-OPS including measured ambient air conditions (to have an idea about the variability of wind speed, direction, air temp, etc.) would be helpful.

Response: I would like to add the information of the environmental variables in supplementary materials, but probably not in the main manuscript in order to simply figure 7 in this paper. Please see supplementary figure 1.

4. Page 4, line 24: How often did you acquire single beam spectra (how many spectra did you measure during the operational period - also to make the number of 877 valid OP-FTIR spectra more valuable)?

Response: Gas sample were continuously collected from the S-OPS, and the collected gas samples were measured for N2O concentrations every ten seconds using the difference frequency generation (DFG) mid-IR laser-based N2O/H2O analyzer (IRIS 4600). Then, the measured N2O concentrations were averaged every thirty minutes to represent the 'actual' ambient N2O concentrations at the 30-min interval. The 30-min averaged N2O was used to benchmark the concentrations derived from the OP-FTIR spectrum. A single beam spectrum was collected every minute based on 64 sample scans. Within a 30-min interval, two to three single beam spectra were collected and measured N2O concentrations. These two-three 'one-minute' N2O concentrations were also averaged to calculate 30-min N2O, compared with the SOPS-measured concentrations. An example was shown in supplementary table 1. The whole OP-FTIR spectra used in this study should be 793 spectra.

5. Page 5, line 17: What do you mean here: "we selected ninety spectra containing 338 ppbv N2O and ninety-three spectra containing 400 ppmv CO2?" These spectra do contain the same concentration like the measurements at the S-OPS? Is the impact of IR absorbance due to water vapor within this spectra not so significant (my interpretation of figure 2d)? (In Figure 2 the readers find the histograms of 418 half-hour

average-intervals? In Fig.2a the light grey line is located at x = 340 ppb and not at x = 338 ppb, by the way ...).

Response: That is correct. Concentrations of N2O (338 ppbv) and CO2 (400 ppmv) were measured by the S-OPS. The OP-FTIR and S-OPS were deployed at the same path and used to collect OP-FTIR spectra and measure N2O/CO2 concentrations, respectively. Ninety spectra containing 338 ppbv N2O (measured by the S-OPS) and ninety-three spectra containing 400 ppmv CO2 (also measured by the S-OPS) were used to test the performances of quantification methods, including least square models, SB backgrounds usages, and spectral windows. 'Is the impact of IR absorbance due to water vapor within this spectra not so significant?': Water vapor content was measured from 0.4 to 2.0 %, and ambient temperature was measured from 10 to 35 °C from June 9-20th 2014. One of the objectives of this study is to investigate the sensitivity of the OP-FTIR to ambient water vapor and temperature. Since spectra containing consistent concentrations of N2O and CO2 but the varied water vapor content and temperature, the variations in the FTIR-calculated concentrations (Fig.5 and 6) resulted from the confounding effects of ambient water vapor interferences and temperature. I will modify the position of the light grey line from 340 ppbv to 338 ppbv before the submission.

6. Page 6, line 23: I see the potential and limitation of the here discussed methods to obtain a target gas free background spectrum. In my opinion, using one background spectrum per day for the zap-bkg determination is to less due to the extensive impact of changing environmental conditions on the measured IR spectra (which is surely occurring during the day in ambient air conditions like air humidity, pressure variability, ....).

Response: Ideally, the zap-bkg spectrum needed to be created from each sample single beam spectrum to obtain absorbance. In this study, however, we only created one zap-bkg spectrum for each day. This idea was inspired by the methodology of the zero-path single beam spectrum used for absorbance spectra conversion. For

continuous gas measurements, zero-path single beam spectrum was usually acquired once per day and used to convert sample single beam spectra to absorbance. Also, it is time-consuming to create the zap-bkg spectrum for each single beam spectrum for continuous gas measurements. Yes, I would agree that this is the potential limitation of using the zap-bkg method and suggest that the future OP-FTIR users can test the effect of multiple zap-bkg on gas quantification before continuous gas measurements.

7. Page 6, line 30: what is meant with: all data points are stored as one data file? You calculated from each SB spectra the synthetic background and store this in the same manner like the original data spectra and use these files for the calculation of abs-spectra? How many data points are used to determine the smooth background spectra function (which order of polynom).

Response: The idea of the synthetic SB background is to select multiple points from the sampled SB spectrum and these selected points were used to fit the curvature of the SB spectrum using the polynomial function. The positions (or wavenumbers) of these selected points are important and need to be consistent for every sample SB spectrum. For instance, the positions must not be selected within the absorbance features. These data points were selected from a quality SB spectrum acquired from each day, and the positions of these data points were stored in one file (a data-point file is a feature provided by the IMACC software). A data-point file was applied in sample SB spectra to make sure the points applied in every sample SB spectrum have consistent positions. Six points within 2050-2500 cm-1 were selected from the SB spectrum to smooth the SB background spectrum using polynomial function (six orders) (Figure 3b in the paper) for N2O and CO2 quantification.

8. Page 8, line 27: humidity Response: Yes, I will change to the humidity.

9. Page 9 / 10: Figure 5 and 6 imply, that the authors did not show all the results. To evaluate the presented methods in order to agree with the proposed "optimal approaches", a comparison of all concentration estimations (CLS, PLS, used spectral

[Figure]

**[AMTD]{.underline}**

Interactive
comment

windows and background spectra) should be shown (or at least mentioned for instance as a table). Otherwise, the assessment is very hard for the reader... (For instance in chapter 3.3 the PLS for CO2 is missing.)

Response: Yes, that would be helpful for readers to follow the approaches, including CLS, PLS, SB backgrounds, and spectral windows, by using tables. We did not analyze CO2 concentrations using the PLS model because we are limited to acquire the wet CO2 reference spectra (CO2/H2O mixed spectra). Therefore, the 'optimal approach' might imply that the CLS model is better than the PLS model for CO2 quantification and mislead readers. I think that would be a good idea to make a short statement that CO2 was only determined by the CLS model but different processes (e.g., SB background spectra, spectral windows).

[Figure]

**Fig. 1.** Supplementary figure 1: Measurements of N2O/CO2 concentrations (S-OPS and OP-FTIR), ambient wind speed and direction, temperature, and humidity from June 9th to 20th 2014

An example of the timing for measuring $N_2O$ concentrations using the S-OPS (SOPS-$N_2O$) and for acquiring the OP-FTIR single beam (SB) spectra (e.g., FTIR-$N_2O$) on June 9[th], 2014. The measurement period was from June 9-20[th] 2014, and the total 793 SB spectra were collected by the OP-FTIR.

| Time of SOPS-$N_2O$ (UTC) | Time of FTIR-$N_2O$ (UTC) | Time of SOPS-$N_2O$ (UTC) | Time of FTIR-$N_2O$ (UTC) |
|---|---|---|---|
| 06/09/2014 16:00 | 06/09/2014 15:37 | 06/09/2014 22:00 | 06/09/2014 21:33 |
|  | 06/09/2014 15:39 |  | 06/09/2014 21:58 |
| 06/09/2014 16:30 | 06/09/2014 16:02 |  | 06/09/2014 21:59 |
|  | 06/09/2014 16:03 | 06/09/2014 22:30 | 06/09/2014 22:22 |
|  | 06/09/2014 16:28 |  | 06/09/2014 22:24 |
| 06/09/2014 17:00 | 06/09/2014 16:53 | 06/09/2014 23:00 | 06/09/2014 22:49 |
|  | 06/09/2014 16:55 | 06/09/2014 23:30 | 06/09/2014 23:13 |
| 06/09/2014 17:30 | 06/09/2014 17:18 |  | 06/09/2014 23:15 |
|  | 06/09/2014 17:19 | 06/10/2014 00:00 | 06/09/2014 23:40 |
| 06/09/2014 18:00 | 06/09/2014 17:44 | 06/10/2014 00:30 | 06/10/2014 00:05 |
| 06/09/2014 18:30 | 06/09/2014 18:09 |  | 06/10/2014 00:29 |
|  | 06/09/2014 18:10 | 06/10/2014 01:00 | 06/10/2014 00:31 |
| 06/09/2014 19:00 | 06/09/2014 18:35 |  | 06/10/2014 00:56 |
| 06/09/2014 19:30 | 06/09/2014 19:00 | 06/10/2014 01:30 | 06/10/2014 01:20 |
|  | 06/09/2014 19:01 | 06/10/2014 02:00 | 06/10/2014 01:45 |
|  | 06/09/2014 19:25 |  | 06/10/2014 01:47 |
|  | 06/09/2014 19:26 | 06/10/2014 02:30 | 06/10/2014 02:12 |
| 06/09/2014 20:00 | 06/09/2014 19:51 | 06/10/2014 03:00 | 06/10/2014 02:36 |
| 06/09/2014 20:30 | 06/09/2014 20:16 |  | 06/10/2014 02:38 |
|  | 06/09/2014 20:17 | 06/10/2014 03:30 | 06/10/2014 03:01 |
| 06/09/2014 21:00 | 06/09/2014 20:42 |  | 06/10/2014 03:03 |
| 06/09/2014 21:30 | 06/09/2014 21:07 |  | 06/10/2014 03:28 |
|  | 06/09/2014 21:08 | 06/10/2014 04:00 | 06/10/2014 03:53 |

**Fig. 2.** Supplementary table 1

---

## Author Response (AR1)

**Author's Response**

**Part-I: A point-by-point response**

These responses were to integrate comments from the public referees and further explanations from authors.

5    1.   Title: "...GHG concentrations from agricultural soils" - you measure GHG concentrations in ambient air above ground surface - these concentrations can stem from emissions from the soil (including roots and microorganisms) or from the above ground vegetation. You do not measure fluxes - so be careful within your introduction - to estimate fluxes the concentration is only one of the variables needed!!!

10          **Response**: I would modify this title to 'Application of Open Path Fourier Transform Infrared Spectroscopy (OP-FTIR) to Measure Greenhouse Gas Concentrations from Agricultural Fields'

    2.   Page 2, line 10 ff: The authors mention chamber measurements as the most common way to investigate emissions from soils. In the same time they point out the relatively small footprint as the main limit of this method. However,

15       I would like to introduce in this context the opportunity to measure GHG fluxes on larger scale using the Eddy Covariance flux measurements. This method is also an established method nowadays and a common micro-meteorological technique with an increased footprint to determine emissions for instance of $CO_2$, $CH_4$ and water from soils and vegetation. (e.g., Baldocchi, D. (2003): Assessing the eddy covariance technique for evaluating carbon dioxide exchange rates of ecosystems: past, present and future. https://doi.org/10.1046/j.1365-

20       2486.2003.00629.x). Concerning to comment No. 1 - all methods are based on the measurement of the concentrations of GHGs and have their own processes to obtain emission rates.

          **Response**: It is a good idea to point out the eddy covariance and its advantage (e.g., larger footprint) for gas flux measurements. I would briefly introduce this method into the context. Chamber measurements are the most

25       common method for soil gas emission measurements because this method also provides numbers of strengths. One of the advantages is that chambers are sensitive enough to make comparisons of gas emissions in different treatments (e.g., field and N management practices in small plots (< 3 ha)), which is challenging to the eddy covariance. The OP-FTIR combined with inversion dispersion techniques (e.g., backward Lagrangian stochastic dispersion model) is capable of measuring gas emissions frequently and with a field-scale footprint (1-3 ha), that

30       can both compensate the limitations of chamber measurements and measure gas emissions from different treatment plots. That is the reason why eddy covariance was not introduced in the context in the first place.

    3.   In section 2.3 (Page 5, line 17), two anonymous referees suggested further explaining the rationality of "Ninety spectra containing $338\pm0.3$ ppbv $N_2O$ and ninety-three spectra containing $400\pm3.0$ ppmv $CO_2$ were selected from

35       these valid spectra to calculate concentrations of $N_2O$ and $CO_2$, respectively, using different quantitative methods."

**Response**: The S-OPS and OP-FTIR were deployed at the same path to measure the path-averaged $N_2O/CO_2$ concentrations and collect OP-FTIR spectra simultaneously. Ninety FTIR spectra containing $N_2O$ concentrations from 337.4 – 338.5 ppbv (measured by the S-OPS), and ninety-three spectra containing $CO_2$ concentrations from 405.4 – 394.9 ppmv (also measured by the S-OPS) were collected (see the following table) and quantified for $N_2O/CO_2$ concentrations using different quantification methods (including least square models, single beam (SB) backgrounds, and spectral windows). These spectra, containing consistent $N_2O/CO_2$ concentrations but different water vapour content and temperature, can be used to examine the sensitivity of quantification methods to water vapour and temperature. For instance, the variations in the FTIR-calculated concentrations (figure--5 and -6 shown in the manuscript) resulted from the confounding effects of ambient water vapor interferences and temperature.

Table - The S-OPS measured concentrations of $N_2O/CO_2$ used for OP-FTIR quantitative method evaluations.

| Gases | Statistics of the S-OPS measured concentrations | | | | | |
|---|---|---|---|---|---|---|
| | Mean | SD | Max | Median | Min | n |
| $N_2O$ (ppbv) | 337.9 | 0.3 | 338.5 | 337.9 | 337.4 | 90 |
| $CO_2$ (ppmv) | 399.8 | 3.0 | 405.4 | 398.9 | 394.9 | 93 |

4. In section 2.3: How often did you acquire a single beam spectrum? (how many spectra did you measure during the operational period - also to make the number of 877 valid OP-FTIR spectra more valuable)?

**Response**: Gas samples were continuously collected from the S-OPS and measured for $N_2O$ concentrations every 10 s using the difference frequency generation (DFG) mid-IR laser-based $N_2O/H_2O$ analyzer (IRIS 4600). The measured $N_2O$ concentrations were averaged every 30 min to represent the 'actual' ambient $N_2O$ concentrations at the 30-min interval. Within the 30-min interval, the environment (e.g., gas concentrations, water vapour content, and temperature) was assumed stationary, meaning that the averaged quantities of variables were invariant. The 30-min averaged $N_2O$ was used to benchmark the concentrations derived from the OP-FTIR spectra. A SB spectrum was collected every minute based on 64 sample scans. Within a 30-min interval, two to three SB spectra were collected and measured $N_2O$ concentrations. The sampling time was shown in the following table. These two-three 'one-minute' $N_2O$ concentrations were also averaged to calculate 30-min averaged $N_2O$ to compare with the SOPS-measured concentrations. The whole OP-FTIR spectra used in this study should be 793 spectra rather than 877.

Table – An example of the timing for measuring $N_2O$ concentrations using the S-OPS (SOPS-N2O) and for acquiring the OP-FTIR single beam (SB) spectra (FTIR-N2O) on 9 Jun. 2014

| Time of SOPS-N$_2$O (UTC) | Time of FTIR-N$_2$O (UTC) | Time of SOPS-N$_2$O (UTC) | Time of FTIR-N$_2$O (UTC) |
|---|---|---|---|
| 06/09/2014 16:00 | 06/09/2014 15:37 | 06/09/2014 22:00 | 06/09/2014 21:33 |
|  | 06/09/2014 15:39 |  | 06/09/2014 21:58 |
| 06/09/2014 16:30 | 06/09/2014 16:02 |  | 06/09/2014 21:59 |
|  | 06/09/2014 16:03 | 06/09/2014 22:30 | 06/09/2014 22:22 |
|  | 06/09/2014 16:28 |  | 06/09/2014 22:24 |
| 06/09/2014 17:00 | 06/09/2014 16:53 | 06/09/2014 23:00 | 06/09/2014 22:49 |
|  | 06/09/2014 16:55 | 06/09/2014 23:30 | 06/09/2014 23:13 |
| 06/09/2014 17:30 | 06/09/2014 17:18 |  | 06/09/2014 23:15 |
|  | 06/09/2014 17:19 | 06/10/2014 00:00 | 06/09/2014 23:40 |
| 06/09/2014 18:00 | 06/09/2014 17:44 | 06/10/2014 00:30 | 06/10/2014 00:05 |
| 06/09/2014 18:30 | 06/09/2014 18:09 |  | 06/10/2014 00:29 |
|  | 06/09/2014 18:10 | 06/10/2014 01:00 | 06/10/2014 00:31 |
| 06/09/2014 19:00 | 06/09/2014 18:35 |  | 06/10/2014 00:56 |
| 06/09/2014 19:30 | 06/09/2014 19:00 | 06/10/2014 01:30 | 06/10/2014 01:20 |
|  | 06/09/2014 19:01 | 06/10/2014 02:00 | 06/10/2014 01:45 |
|  | 06/09/2014 19:25 |  | 06/10/2014 01:47 |
|  | 06/09/2014 19:26 | 06/10/2014 02:30 | 06/10/2014 02:12 |
| 06/09/2014 20:00 | 06/09/2014 19:51 | 06/10/2014 03:00 | 06/10/2014 02:36 |
| 06/09/2014 20:30 | 06/09/2014 20:16 |  | 06/10/2014 02:38 |
|  | 06/09/2014 20:17 | 06/10/2014 03:30 | 06/10/2014 03:01 |
| 06/09/2014 21:00 | 06/09/2014 20:42 |  | 06/10/2014 03:03 |
| 06/09/2014 21:30 | 06/09/2014 21:07 |  | 06/10/2014 03:28 |
|  | 06/09/2014 21:08 | 06/10/2014 04:00 | 06/10/2014 03:53 |

5. In section 2.4, the authors stated that "each sampled spectrum was acquired by co-adding 64 single-sided interferograms (IFGs) at the nominal resolution of 0.5 cm$^{-1}$, which accounted for 32,000 data points were collected with an interval of 0.241 wavenumbers between data points, …" For interferograms, the unit of the interval of data points is cm, not wavenumber.

**Response**: This sentence might be confused. It means that a resolution of 0.5 cm$^{-1}$ accounting for a data point every 0.241 wavenumbers was used for acquiring SB spectra (400 – 4000 cm$^{-1}$), and approximate 32,000 data points were in the interferogram.

6. In section 2.4, the authors employed some criteria to remove low-quality IFGs, which includes those of very intense centerburst. It is true that intense-centerburst IFGs result in severe non-linear response of MCT detectors. However, such IFGs have high signal-to-noise ratio, and would be valid once the non-linear detector response is corrected. The authors might be interested in correcting method (L. Shao, P.R. Griffiths, Anal. Chem., 2008, 80(13), 5219).

**Response**: The maximum A/DC capacity in this study was approximately 2.49 Volts. The optical path length of the OP-FTIR was 300 meters. The maximum or minimum value of the IFG centerburst in this study located between

0.61-1.14 volts, which prevented the MCT detector from saturation as well as avoided the non-linear response of the detector.

7. In section 2.5.1 (Page 6, line 23): I see the potential and limitation of the here discussed methods to obtain a target gas free background spectrum. In my opinion, using one background spectrum per day for the zap-bkg determination is to less due to the extensive impact of changing environmental conditions on the measured IR spectra (which is surely occurring during the day in ambient air conditions like air humidity, pressure variability, ....).

**Response**: Ideally, the zap-bkg spectrum needed to be created from each sample SB spectrum to obtain absorbance. In this study, however, we only created one zap-bkg spectrum for each day. This idea was inspired by the methodology of the zero-path SB spectrum used for absorbance spectra conversion. For continuous gas measurements, zero-path SB spectrum was usually acquired once per day and used to convert sample SB spectra to absorbance. Also, it is time-consuming to create the zap-bkg spectrum for each SB spectrum for continuous gas measurements. Yes, I would agree that this is the potential limitation of using the zap-bkg method and suggest that the future OP-FTIR users can test the effect of multiple zap-bkg on gas quantification before continuous gas measurements.

8. In section 2.5.1, the authors stated using "a high-order fitting function" as the synthetic background. It is better to be specific about the function, is it a quadratic, cubic polynomial, or something else?

**Response**: Numerous data points were selected from the field SB spectrum. A polynomial function was used to fit the field spectrum to synthesize the SB background without features of gas absorption.

9. In section 2.5.1 (Page 6, line 30): what is meant with: all data points are stored as one data file? You calculated from each SB spectra the synthetic background and store this in the same manner like the original data spectra and use these files for the calculation of abs-spectra? How many data points are used to determine the smooth background spectra function (which order of polynom).

**Response**: The idea of the synthetic SB background is to select multiple points from the sampled SB spectrum, and these selected points were used to fit the curvature of the SB spectrum using the polynomial function. The positions (or wavenumbers) of these selected points are important and need to be consistent for every sample SB spectrum. For instance, the positions must not be selected within the absorbance features. These data points were selected from a quality SB spectrum acquired from each day, and the wavenumbers of these selected data points were stored in one file (a data-point file is a feature provided by the IMACC software). A data-point file was applied in sample SB spectra to make sure the points applied in every sample SB spectrum have consistent positions. Six points

within 2050-2500 cm$^{-1}$ were selected from the SB spectrum to smooth the SB background spectrum using polynomial function (six orders) (Figure 3b in the paper) for $N_2O$ and $CO_2$ quantification.

10. In section 2.5.2, some useful information about PLS models is not provided, such as the number of calibration spectra (to build the model), the concentration range that the model covers, the number of factors for the model.

**Response**: Sixty mixed-gas (i.e., $N_2O$ + water vapor) spectra were collected from the lab-based FTIR joined with the multi-pass gas cell (the optical path length of 33 meters). Concentrations of $N_2O$ and water vapor ranged from 0.3 - 0.7 ppmv and 7000 – 30,000 ppmv, respectively. More details of the calibration spectra were shown in the supplementary table-2.

11. In section 2.6, it is better to be specific about the statistical tests, are they t-test or paired t-test?

**Response**: For $N_2O$ analysis, twelve quantitative models used for $N_2O$ concentration calculations from ninety OP-FTIR spectra were examined to optimize the combinations of SB backgrounds (i.e., zap- and syn-bkg), multivariate models (i.e., CLS and PLS), and analytical windows (i.e., $W_N1$-$W_N4$). In order to compare the means of the twelve populations, the Fisher's Least significant difference (LSD) was used for multiple comparisons ($\alpha = 0.05$). Likewise, the LSD was also used to compare six population means for the $CO_2$ analysis.

12. In section 3.2, the authors present the result of CLS (zap-bkg) and CLS (syn-bkg), and the result of PLS (syn-bkg). Why is the result of PLS (zap-bkg) absent? It seems that the authors did not apply PLS to estimate the concentrations of $CO_2$, as they did in case of $N_2O$. The reason should be explained.

**Response**: The syn-bkg is one of the recommended methods for converting the SB to absorbance spectra, but the zap-bkg was the newly proposed method. Thus, the syn-bkg was used to examine the feasibility and the performance of the zap-bkg. The identical field SB spectra, analytical windows, and CLS model were used to calculate gas concentrations from the absorbance spectra converted by zap- and syn-bkg. For both $N_2O$ and $CO_2$ analyses, the zap-bkg method led to more biases in concentration calculations than the syn-bkg using CLS models. For instance, the zap-bkg underestimated $N_2O$ concentrations by a least 9%, and the syn-bkg improved the quantitative accuracy (Fig. 5 shown in the manuscript). Simply removing the gas absorption feature by the linear function potentially resulted in biases. Thus, applying the PLS to quantify gas concentration from the spectra converted by zap-bkg is unlikely to improve the quantitative accuracy. For simplification, the results of the integrated uses of the zap-bkg and PLS model were not reported. Also, since we were limited to acquire wet $CO_2$ reference spectra ($CO_2/H_2O$ mixing spectra), the PLS model was not used for $CO_2$ quantification.

13. General comment: Time series on GHG concentrations at S-OPS including measured ambient air conditions (to have an idea about the variability of wind speed, direction, air temp, etc.) would be helpful.

**Response**: Information of the environmental variables was updated in supplementary materials (supplementary figure 1) and in figure-7.

14. Fig.7(b) is strange. As stated in the Fig.7(b), bias = FTIR – S-OPS. According to this formula, the bias between 11/6/2014 and 12/6/2014 is negative, since the FTIR concentrations are clearly lower than S-OPS. But in the figure the corresponding bias is positive.

**Response**: The right Y-axis of bias (%) of figure-7 is reverse, so the biases should be negative. The concentrations of $N_2O$ (ppbv) and $CO_2$ (ppmv) measured by the S-OPS and OP-FTIR spectra needed to be corrected by the humidity content in the air (dry air correction). The original Figure-7 showed the dry-air corrected concentrations of $N_2O$ and $CO_2$; however, $CO_2$ concentrations calculated from OP-FTIR were not corrected by humidity content by accident. The new Figure-7 with the dry-air corrected $CO_2$ concentrations (FTIR-$CO_2$) were updated in the manuscript.

**Part-II: A list of relevant changes**

Summary – Some unclear or ambiguous sentences and paragraphs were rephrased to improve understandings of this paper. The marked-up version of this article was attached (Part-III), and the list of the important changes is as follows:

1. **Article title**: The title was changed to 'Application of Open Path Fourier Transform Infrared Spectroscopy (OP-FTIR) to Measure Greenhouse Gas Concentrations from Agricultural Fields'.

2. **Affiliations**: The 'Department of Earth, Atmospheric and Planetary Sciences, Purdue University, West Lafayette, IN 47907' was added to one of Cliff T. Johnston's affiliations.

3. **Abstract (page 1, line 12)**: Changed the sentence of '… measure concentrations of greenhouse gases (e.g., $N_2O$ and $CO_2$) emitted from agricultural soils …' to '… measure concentrations of $N_2O$ and $CO_2$ at a maize cropping system …".

4. **Introduction (Page 2, line 19)**: Briefly introduced the eddy covariance based on suggestions from the Anonymous Referee #3 by adding sentences of 'It is worth mentioning that the eddy covariance flux measurement method, one of the most common micro-meteorological techniques used to investigate gas exchanges in the agroecosystem, is capable of measuring gas fluxes frequently with an increased footprint (Baldocchi, 2003). A large-scale flux measurement (hundred meters to several kilometres) using this method, however, make comparisons among field-scale treatments (1-5 ha) more difficult than chamber methods (Schmid, 1994; Denmead, 2008).'

5. **Materials and experimental methods (Page 5, line 4)**: Rephrased the original sentence to 'Gas samples were drawn through an S-OPS line by a sampling pump in the GSS at approximately 7 L·min$^{-1}$ and collected into a

Teflon® ambient pressure chamber. Then, $N_2O$ and $CO_2$ analysers drew air samples from the ambient pressure chamber to measure the 'actual' path-averaged concentrations of $N_2O$ and $CO_2$ along the OP-FTIR path (Heber et al., 2006). The measured $N_2O/CO_2$ concentrations were used to benchmark concentrations calculated from the OP-FTIR spectrum.'

6. **Materials and experimental methods (Page 5, line 22)**: Rephrased the original paragraph of the 2.3 section of 'Overview of ambient temperature and concentrations of $N_2O/CO_2$/water vapor'.

7. **Materials and experimental methods (Page 5, line 30)**: Rephrased the original sentence to 'A spectra range of 500.0-4000.0 $cm^{-1}$ and a resolution of 0.5 $cm^{-1}$ were selected for spectra acquisition. Each sampled spectrum was acquired by co-adding 64 single-sided interferograms (IFGs) using the AutoQuant Pro4.0 software package (MIDAC Corporation, Irvine, CA).'

8. **Materials and experimental methods (Page 6, line 32)**: Added the sentence of 'Six points within 2050.0-2500.0 $cm^{-1}$ were selected to fit the curvature of the SB spectrum using a polynomial function to create a syn-bkg SB spectrum (Fig. 3b)' to clarify the process used to generate the 'synthetic' SB background in this study, which is also based on suggestions from the Anonymous Referee #2 and #3.

9. **Results and discussion (Page 8)**: Sections-3.2 (Nitrous oxide (338 ppbv)), the first paragraph in section-3.3 (Carbon dioxide (400 ppmv)), and the first two sentences in section-3.4 (Diurnal $N_2O/CO_2$ estimations) (page 10, line 27-30) were rephrased to improve sentence fluency and clarify ambiguous concepts. Details were shown in the marked-up manuscript.

10. **Figure-2 (Page 17):** The gray bar shown in Figure 2a was aligned to 338±0.3 ppbv, which was suggested by the Anonymous Referee #3.

11. **Figure-7 (Page 22)**: Two changes shown in Figure 7 were (1) the 30-min averages of air temperature and wind speed measured from 9-19 June 2014 were added in the figure, and (2) the $CO_2$ concentrations measured from the OP-FTIR spectra were not calibrated by the humidity content in the air (dry air correction) in the original figure-7. The dry-air corrected $CO_2$ concentrations were updated to the new figure-7. This issue was addressed by C-H Lin on 13 Dec. 2018 on the AMTD forum.

**Part-III: A marked-up manuscript version**

[revised manuscript text omitted]